# Value-Guided Decision Transformer: A Unified Reinforcement Learning Framework for Online and Offline Settings

Hongling Zheng[1]    Li Shen[2][†]    Yong Luo[1][†]    Deheng Ye[3]    Shuhan Xu[1]    Bo Du[1]
Jialie Shen[4]    Dacheng Tao[5]

[1] School of Computer Science, National Engineering Research Center of Multimedia Software
and Hubei Key Laboratory of Multimedia and Network Communication Engineering,
Wuhan University, Wuhan, China
[2]Shenzhen Campus of Sun Yat-sen University, Shenzhen, China    [3]Tencent Inc., Shenzhen, China
[4]City St George's, University of London, UK    [5]Nanyang Technological University, Singapore
{hlzheng, luoyong, xushuhan, dubo}@whu.edu.cn   dericye@tencent.com
{mathshenli, dacheng.tao}@gmail.com   jerry.shen@citystgeorges.ac.uk

## Abstract

The Conditional Sequence Modeling (CSM) paradigm, benefiting from the transformer's powerful distribution modeling capabilities, has demonstrated considerable promise in Reinforcement Learning (RL) tasks. However, much of the work has focused on applying CSM to single online or offline settings, with the general architecture rarely explored. Additionally, existing methods primarily focus on deterministic trajectory modeling, overlooking the randomness of state transitions and the diversity of future trajectory distributions. Fortunately, value-based methods offer a viable solution for CSM, further bridging the potential gap between offline and online RL. In this paper, we propose Value-Guided Decision Transformer (VDT), which leverages value functions to perform advantage-weighting and behavior regularization on the Decision Transformer (DT), guiding the policy toward upper-bound optimal decisions during the offline training phase. In the online tuning phase, VDT further integrates value-based policy improvement with behavior cloning under the CSM architecture through limited interaction and data collection, achieving performance improvement within minimal timesteps. The predictive capability of value functions for future returns is also incorporated into the sampling process. Our method achieves competitive performance on various standard RL benchmarks, providing a feasible solution for developing CSM architectures in general scenarios. Code is available at here.

## 1 Introduction

Offline reinforcement learning (Offline RL) [1] aims to develop a reward-maximizing RL strategy using offline data. This approach is highly valuable in real-world scenarios where online data collection is expensive, time-consuming, or impractical. Transformer [2] is widely regarded for its capacity to capture complex data distributions and long-term temporal dependencies, becoming a foundational architecture in fields such as Natural Language Processing [3, 4] and Computer Vision [5, 6]. Inspired by this success, Decision Transformer (DT) [7] and its variants [8, 9] introduce the transformer to the field of offline RL, demonstrating its powerful capabilities in Conditional Sequence Modeling (CSM) [10]. Specifically, DT integrates cumulative rewards, states, and actions

---

[†]Corresponding authors.

39th Conference on Neural Information Processing Systems (NeurIPS 2025).

into a tuple and trains on offline datasets autoregressively to output appropriate actions. This approach relaxes the MDP assumption by considering multiple historical steps, allowing the model to handle long sequences and avoid stability issues associated with bootstrapping [11].

The CSM method is essentially goal-conditioned behavior cloning, which optimistically treats the highly incidental performance gains resulting from the randomness in offline data as a general expectation. As a result, while it performs well in deterministic environments, it struggles to achieve good performance in stochastic environments or when faced with suboptimal data. Some works [12, 13] have combined value functions trained on offline data with CSM to generate suboptimal trajectory stitching, guiding the agent's robust learning. However, these approaches typically introduce the value function by simply re-labeling the return-to-go (RTG) or directly using it as penalty terms, with little consideration given to how to explore further the optimization upper bound of the value function in the CSM setting and effectively integrate it with the DT to curb its overly optimistic behavior cloning.

Bridging the gap between offline and online reinforcement learning remains a central challenge. One promising direction that has emerged in recent years is offline-to-online RL [14, 15]. Within the CSM framework, a commonly used method is the Online Decision Transformer (ODT) [16], which extends DT training into the online phase while maintaining the same supervised learning paradigm used in offline RL. However, ODT faces challenges in achieving expert-level performance in online scenarios with limited or suboptimal data, primarily due to its inability to compose or integrate suboptimal trajectories effectively. Furthermore, ODT demonstrates significant advantages only after online fine-tuning, while its performance in offline scenarios remains suboptimal. There is still limited research on how the CSM architecture can generalize across offline and online settings.

To remedy these drawbacks, we propose the Value-Guided Decision Transformer (VDT), a unified RL framework for online and offline settings. (1) **Offline Training Phase:** Based on the DT training framework, we combine a multi-step Bellman-optimized Q-function and state-value function to explore the upper bound of value estimation. Value guidance is integrated with the behavior cloning of Decision Transformer (DT) through advantage-weighted learning, while the maximum estimated value is concurrently used as a penalty term to regularize the expected value of the current action distribution. The coupling and regularization effectively mitigate the overly optimistic estimates of CSM in stochastic environments and enable trajectory stitching from suboptimal data. (2) **Online Tuning Phase:** VDT refines the value function and the policy through limited interactions. Integrating the trajectory-level replay buffer and RTG alignment achieves significant performance improvements within minimal timesteps. (3) **Sampling Process:** The Q-function evaluates the expected future return of each action the policy generates under different RTGs within a predefined evaluation horizon and selects the optimal decision. The introduction of the value function significantly improves the DT's performance in both pure offline and offline-to-online settings, regardless of data quality or reward sparsity, and further bridges the potential gap between offline and online RL.

The main contributions of this work are as follows:

- We incorporate the value function into the CSM architecture and enhance behavior cloning with advantage-weighted learning and regularization constraints. These components enable VDT to stitch together suboptimal trajectories under value-based guidance and achieve robust performance across varying-quality datasets. We further provide a theoretical guarantee of its superior performance.

- We leverage the inherent strengths of the value function to fine-tune the policy with a limited number of interactions in the online phase. By introducing the trajectory-level replay buffer and return-to-go alignment, we bridge the gap between offline training and online tuning, offering insights into the design of generalizable architectures.

- We demonstrate the effectiveness of VDT across a broad spectrum of benchmarks, exhibiting superior performance in pure offline and offline-to-online settings.

## 2 Related works

**CSM for Offline RL.** In contrast to online RL, offline RL [17] focuses on training models and performing trial-and-error using offline data without environmental interaction to arrive at appropriate strategies. Recently, CSM for RL [18, 19], represented by the transformer architecture, has further

demonstrated the advantages of data-driven policy learning. DT [7] is trained on an offline dataset of triplets encapsulating return-to-go $\hat{r}_t$, state $s_t$, and action $a_t$, and outputs the optimal action. The $\hat{r}_t$ token quantifies the cumulative reward from the current time step to the end of the episode. During training, DT processes a trajectory sequence $\tau_t$ in an auto-regressive manner, which encompasses the most recent $K$-step historical context:

$$\tau_t = (\hat{r}_{t-K+1}, s_{t-K+1}, a_{t-K+1}, \ldots, \hat{r}_t, s_t, a_t) \tag{1}$$

The prediction head associated with a state token $s_t$ is trained to predict the corresponding action $a_t$. Regarding continuous action spaces, the training objective is to minimize the mean-squared loss $L_{DT}$.

$$L_{DT} = \mathbb{E}_{\tau_t \sim \mathcal{D}} \left[ \frac{1}{K} \sum_{t=t-K+1}^{K} (a_t - \pi_\theta (\tau_t)_i)^2 \right] \tag{2}$$

Subsequent work has made various improvements to DT, including prompt tuning [20], trajectory concatenation [21], and value regularization [22]. These approaches often involve more complex modifications of DT to adapt it to specific tasks.

**Offline-to-Online RL.** Offline-to-online RL focuses on fine-tuning a policy pre-trained on offline data using a limited number of online interactions to improve performance and narrow the gap between offline and online learning. Key challenges in this setting include effectively leveraging offline data for initializing the policy and mitigating distributional shifts during the online adaptation process. Early approaches focus on pre-training policies with offline data [23, 24] and use techniques such as balanced sampling [25], adaptive conservatism [15], and actor-critic alignment [26] to stabilize the transition to the online phase. For efficient online fine-tuning, optimistic exploration strategies are employed, utilizing Q-ensembles [27], uncertainty-guided exploration [28], or model-based uncertainty estimation [29, 30]. While most existing methods follow Q-learning paradigms, the use of CSM architectures such as DT for offline-to-online reinforcement learning remains largely underexplored.

**Value-based Offline RL.** The value-based method is one of the most prominent categories for addressing the distribution shift problem in offline RL. Primarily previous works generally address this problem in one of three ways: (1) constraining the learned policy to the behavior policy [23, 31]; (2) constraining the learned policy by making conservative estimates of future rewards [24, 22]; (3) introducing model-based methods, which learn a model of the environment dynamics to generate more data for policy training and perform pessimistic planning in the learned MDP [32, 33].

The most relevant work to ours is ODT [16], the first to establish a pipeline for transitioning from offline to online RL within the DT framework. However, since ODT adopts a strategy that mirrors its offline training phase, it struggles to effectively handle suboptimal data and adapt to dynamic or stochastic environments. Moreover, as a few-shot method primarily focused on online fine-tuning, ODT performs significantly worse than most baselines in offline settings. TD3+ODT [34], which is primarily based on the ODT algorithm pipeline, further incorporates the RL gradient from the critic as an additional penalty term in the loss function. Our method is distinguished by the effective optimization of the value function and its integration with the DT, enabling the policy to achieve robust performance across varying data quality in offline and online settings. In addition, the evaluation mechanism we design during the sampling process further encourages exploration, facilitating efficient trajectory stitching.

## 3 Preliminary

**Markov Decision Process.** Reinforcement Learning is typically formulated as a Markov Decision Process (MDP), defined by a tuple $(\mathcal{S}, \mathcal{A}, P, r, \gamma)$, where $\mathcal{S}$ represents the state space, $\mathcal{A}$ is the action space, $P$ is the transition function, $r$ is the reward function, and $\gamma \in [0, 1]$ is the discount factor.

In offline RL, the objective is to optimize the RL policy using a previously collected dataset $D = \{(s_t^i, a_t^i, r_t^i, s_{t+1}^i)\}_{i=0}^{N-1}$, consisting of $N$ trajectories. The key distinction in offline RL is that the agent does not interact directly with the environment during the learning phase but instead learns from the historical data. This offline dataset represents the state-action-reward-state transitions the agent experienced during earlier episodes. The agent's task is to extract useful information from these trajectories to improve its decision-making policy without further environmental exploration.

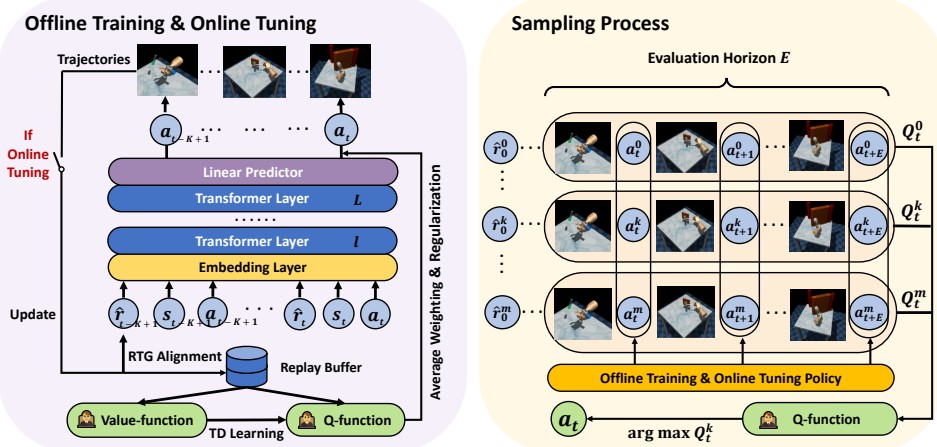

Figure 1: The architecture of Value-Guided Decision Transformer. **Left: Offline Training & Online Tuning.** VDT is trained offline under value guidance and interacts with the environment online to generate trajectories for updating the replay buffer. These trajectories are then used to tune VDT further. **Right: Sampling Process.** At a specific timestep $t$, the policy generates candidate actions within the evaluation horizon $E$ under predefined RTGs, and evaluates them using the Q-function to obtain the optimal action $a_t$.

**Online Decision Transformer.** Online Decision Transformer (ODT) operates in two stages: *offline pretraining* and *online finetuning*. In the offline phase, the model learns from static trajectories using supervised learning, following the standard DT paradigm. In the online phase, it collects new trajectories and refines the policy with supervised updates. During rollouts, the action $a_t$ at timestep $t$ is either computed deterministically as $a_t = \mu_{\text{DT}}(\tau_t)$, or sampled stochastically from the policy $a_t \sim \pi_{\text{DT}}(a_t \mid \tau_t)$. The policy is initialized using the offline dataset and is updated during finetuning as newly collected online trajectories gradually replace the old data buffer.

## 4 Method

This section provides a comprehensive description of the proposed VDT. We break down VDT into three parts: offline training, online tuning, and the sampling process. The **offline training** details how the value function is integrated into the conventional CSM architecture using advantage-weighted learning and regularization terms. The **online tuning** focuses on the collection and processing of online data. Although the model has already incorporated the value guidance during training, due to the inherent randomness of the environment, we perform value evaluation on the predicted actions during the **sampling process** to maximize decision-making performance.

### 4.1 Offline Training

We incorporate value guidance into the DT architecture to enable the CSM architecture to perform robustly in highly stochastic environments and with suboptimal trajectories, and to achieve effective trajectory stitching. As a first step, we learn the Q-function. Specifically, we aim to estimate an upper bound of the Q-function within the support of the dataset's action distribution. We implement the Q-networks $Q_{\theta_1}(s, a)$ and $Q_{\theta_2}(s, a)$, along with their corresponding target networks $Q_{\hat{\theta}_1}(s, a)$ and $Q_{\hat{\theta}_2}(s, a)$, as multi-layer perceptrons (MLPs) with two hidden layers. This architecture provides sufficient representational capacity to approximate complex value functions while maintaining stability during offline training. Inspired by Implicit Q-Learning (IQL) [24], we adopt an expectancy regression technique, which leverages an asymmetric L2 loss to provide an unbiased estimation of the conditional expectile. For a given threshold $\epsilon > 0.5$, this loss function adaptively emphasizes high-return actions by down-weighting the influence of actions with returns below the threshold, thereby approximating the conditional upper bound of the Q-function. To support this, we use an independent state-value function $V_\phi(s)$, implemented as a simple two-layer MLP, and optimize it

using the following objective:

$$L_V(\psi) = \mathbb{E}_{(s_t,a_t)\sim\mathcal{D}}\left[L_2^\epsilon\left(Q_{\hat{\theta}}(s_t,a_t) - V_\psi(s_t)\right)\right],\tag{3}$$

where $L_2^\epsilon(u) = |\epsilon - \mathbb{I}(u < 0)|u^2$. After obtaining the $V_\psi$ with average properties, given that the input to the DT consists of historical trajectories, we choose the n-step Bellman equation to estimate the double Q-networks, which has been shown to provide better stability in practical training compared to single-step optimization [35]. To update $Q_{\theta_i}, i \in [1, 2]$, we use the following formula:

$$\mathbb{E}_{(s_t,a_t,r_t,...,s_{t+n})\sim\mathcal{D}}\left[\left(\sum_{k=0}^{n-1}\gamma^k r_{t+k} + \gamma^n V_\psi(s_{t+n}) - Q_{\theta_i}(s_t,a_t)\right)^2\right]\tag{4}$$

where $\gamma$ is the discount factor. We modify the TD learning procedure to learn an approximation to the optimal Q-function on an offline dataset. In designing the learning objective, we weigh the DT's trajectory modeling loss by computing the advantage function $\min_{i=1,2} Q_{\hat{\theta}_i}(s_t,a_t) - V_\psi(s_t)$, which helps the model identify high-return action trajectories in the dataset and assign them higher importance. As proposed by [36], this advantage-weighted method performs policy optimization within the action distribution supported by the dataset, stabilizing the learning process. Additionally, to further guide the policy towards high-value directions, we compute the policy's action output at the current state and evaluate its value using the target Q-networks, which is then incorporated into the loss function as a regularization term.

$$\mathbb{E}_{\substack{\tau_t\sim\mathcal{D}\\(s_t,a_t)\sim\tau_t}}\left[\exp(\eta(\min_{i=1,2} Q_{\hat{\theta}_i}(s_t,a_t) - V_\psi(s_t)))\|\pi_{DT}(\tau_t) - a_t\|^2 - \lambda\cdot\min_{i=1,2}Q_{\hat{\theta}_i}(s_t,\pi_{DT}(\tau_t))\right]\tag{5}$$

where $\eta$ is an inverse temperature, which we set to 3. The parameter $\lambda$ is a hyperparameter that balances the weight between the two loss terms. Through this loss design, the combination of advantage weighting and the regularization term optimizes the stability of trajectory modeling while guiding the value function to break through the local optima within the data distribution.

**Theorem 4.1.** *Let $\pi_{DT}^*$ be the optimal policy of Equation 5. For any $s \in S$, we have that $V^{\pi_{DT}^*}(s) \geq V^\beta(s)$ and $\pi^*(\mathbf{a} \mid \mathbf{s}) = 0$ given $\beta(\mathbf{a} \mid \mathbf{s}) = 0$.*

**Theorem 4.2.** *For any initial state distribution $\mu$, we have that $V^{\pi^*}(\mu) - V^{\pi_{DT}^*}(\mu) \leq \frac{2\gamma}{(1-\gamma)^2}\cdot$ $\mathbb{E}_{s\sim d^{\pi^*}}\left[\max_{a\notin\mathcal{A}_D(s)} Q^{\pi^*}(s,a) - \max_{a\in\mathrm{supp}(\beta(\cdot|s))} Q^{\pi^*}(s,a)\right].$*

Building on the theoretical foundations established by Theorem 4.1 and the new upper bound provided by Theorem 4.2, we further confirm the effectiveness of Equation 5. These theoretical insights suggest that a policy guided by the value function is likely to outperform the behavior policy. In particular, advantage-weighting and regularization play a crucial role by prioritizing high-value actions, thus steering the learning process towards optimal returns and ensuring consistent improvement over the baseline behavior policy $\beta$. Complete proofs are provided in Appendix A and Appendix B.

## 4.2 Online Tuning

During the online tuning stage, while following almost the same pipeline as the offline training phase, we also introduce several distinct components to achieve the most significant performance gains with the fewest interaction steps.

**Trajectory-Level Replay Buffer.** In the online tuning phase, we employ a replay buffer similar to that used in ODT [16], where the buffer consists of entire trajectories rather than individual transitions. The replay buffer is initially populated with the trajectories that yield the highest returns in the offline dataset. Each time the policy interacts with the environment, we fully roll out an episode using the current policy, then refresh the replay buffer by adding the collected trajectory in a first-in-first-out manner. Afterwards, we update the policy and proceed with another rollout. We use the two-step sampling procedure to ensure that the sub-trajectories of length $K$ in the replay buffer $T_{replay}$ are sampled uniformly. We first sample a single trajectory with probability proportional to its length, then uniformly sample a sub-trajectory of length $K$. Our sampling strategy is akin to importance sampling for environments with non-negative, dense rewards. In those cases, the length of a trajectory is highly correlated with its return.

**Return-to-go Alignment.** During the offline training phase, the RTG at the current step is accumulated from subsequent rewards, and the policy learns conditioned on this RTG. However, in the online

tuning phase, the policy interacts with the environment in real-time based on a predefined RTG, and the induced RTG may differ from the predefined RTG. This discrepancy leads to a mismatch between the expected and actual returns, further affecting the effectiveness of the value function in guiding the policy. To address this, we modify the RTG token at each step of the rolled-out trajectory using the achieved returns, such that the RTG token at time step $t$ is set as $RTG_t = \sum_{j=t}^{|\tau|} r_j$, where $|\tau|$ denotes the trajectory length and $r_j$ denotes the reward at each step. Specifically, for the last timestep of a trajectory, the RTG token equals the immediate reward obtained by the agent upon trajectory termination, which more accurately reflects the return at that step and helps to align the expected and actual returns. Note that, depending on the properties of the environment, the immediate reward at the final step is not necessarily zero.

## 4.3 Sampling Process

Traditional DT uses multiple RTGs for completely independent evaluations and typically selects the trajectory with the highest cumulative return under a specific RTG as the evaluation result, neglecting the potential guidance from alternative RTGs at each timestep. We incorporate multiple RTGs at each trajectory step to overcome this limitation. The value function evaluates the candidate actions, and the optimal action is chosen as the unified decision across all RTG-guided trajectories at that step. This strategy guarantees a unique evaluation outcome, eliminates manual trajectory selection, and effectively integrates the strengths of all RTGs. Specifically, we predefine $m$ candidate RTGs $(\hat{r}_0^0, \hat{r}_0^1, \ldots, \hat{r}_0^m)$ and maintain $m$ parallel trajectories. At each timestep $t$, the policy simultaneously generates $m$ candidate actions $(a_t^0, a_t^1, \ldots, a_t^m)$, where $a_t^k$ is generated under the guidance of RTG $\hat{r}_0^k$. To evaluate these candidates, we introduce an evaluation horizon $E$. For specific candidate action $a_t^k$, the model autoregressively predicts the subsequent $E$-step trajectory $\tau_t^k = (s_t, a_t^k, s_{t+1}^k, a_{t+1}^k, \ldots, s_{t+E}^k)$ under the corresponding RTG $\hat{r}_0^k$, and then computes the cumulative action-value sum over the horizon:

$$Q_t^k = \sum_{i=0}^{E} \gamma^i \cdot Q(s_{t+i}^k, a_{t+i}^k), \tag{6}$$

where $Q$ denotes the action-value function and $\gamma$ is the discount factor. The optimal action $a_t$ is selected as $a_t = \arg\max_{a_t^k} Q_t^k$. This optimal action is then appended to all $m$ trajectories and used to interact with the environment to obtain the next state and reward shared across the $m$ parallel trajectories. Crucially, the parallel nature of trajectory prediction across candidates ensures computational efficiency—despite evaluating $m$ trajectories, the batched computation on modern GPUs results in latency comparable to single-trajectory inference. By employing a value function to evaluate and select the optimal action, we effectively integrate guidance from different RTGs, achieving optimal decision-making at each step while maintaining constant inference time. We concisely outline the VDT pipeline in Appendix D.

Our sampling procedure shares similarities with both CEM [37] and SfBC [38]. Like CEM, we generate multiple candidate actions at each step, evaluate them with a value function over a short planning horizon, and select the best one—essentially a single-step population-based search. Unlike SfBC, which samples from a behavior policy and selects by Q-value, VDT uses RTGs to guide candidate actions and evaluates them in parallel. This integration of RTG guidance and batch evaluation allows VDT to combine the strengths of population search and candidate selection while maintaining efficient inference.

## 5 Experiment

In this section, we extensively evaluate our proposed Value-Guided Decision Transformer (VDT) using the widely recognized D4RL benchmark [39]. As an integrated framework, VDT focuses on performance in offline and offline-to-online (hereafter referred to as "online" for simplicity, without causing ambiguity) settings in the main experiments to ensure the model's generality. Since the offline and online pipelines are nearly identical, we conduct ablation studies under the offline setting to evaluate the shared components.

**Datasets.** We consider five different domains of tasks in the widely used D4RL benchmark: Gym, Adroit, Kitchen, AntMaze and Maze2D. A detailed introduction to these five environments is presented in Appendix E.

**Baselines.** In the offline traing phase, we compare VDT with representative offline RL algorithms from value-based and CSM methods. For value-based methods, including BEAR [40], BCQ [41], CQL [23], MoRel [42], O-RL [43] and COMBO [32]. For CSM methods, including DT, DD [44], EDAC [45], D-QL [46], MPPI [47], StAR [48], GDT [49] and CGDT [12]. In the online tuning phase, we compare VDT with ODT, IQL [24], AWAC [50], CQL [23] and PDT [9], SAC [51] and TD3+BC [52]. The performance scores for these baseline methods are sourced from the best results published in respective papers or from our runs, ensuring a fair comparison.

**Implementation details.** All experiments are carried out on a server with 8 NVIDIA 3090 GPUs, each with 24GB of memory. The experimental hyperparameter configurations of VDT are shown in Appendix C.

## 5.1 Main Experiment

Table 1: Offline training performance of VDT and state-of-the-art baselines on D4RL tasks. For VDT, results are reported as the mean and standard error of normalized rewards over 30 random rollouts (3 independently trained models with 10 trajectories each), generally showing low variance.

| Dataset | Value-Based Methods | | | | | Conditional Sequence Modeling Methods | | | | | | |
|---|---|---|---|---|---|---|---|---|---|---|---|---|
| **Gym Tasks** | BEAR | BCQ | CQL | IQL | MoRel | BC | DT | StAR | GDT | CGDT | DC | VDT |
| halfcheetah-medium-replay-v2 | 38.6 | 34.8 | 37.5 | **44.1** | 40.2 | 36.6 | 36.6 | 36.8 | 40.5 | 40.4 | 41.3 | 39.4 $_{\pm 2.0}$ |
| hopper-medium-replay-v2 | 33.7 | 31.1 | 95.0 | 92.1 | 93.6 | 18.1 | 82.7 | 29.2 | 85.3 | 93.4 | 94.2 | **96.0**$_{\pm 1.9}$ |
| walker2d-medium-replay-v2 | 19.2 | 13.7 | 77.2 | 73.7 | 49.8 | 32.3 | 79.4 | 39.8 | 77.5 | 78.1 | 76.6 | **82.3** $_{\pm 2.1}$ |
| halfcheetah-medium-v2 | 41.7 | 41.5 | 44.0 | **47.4** | 42.1 | 42.6 | 42.6 | 42.9 | 42.9 | 43.0 | 43.0 | 43.9$_{\pm 0.7}$ |
| hopper-medium-v2 | 52.1 | 65.1 | 58.5 | 63.8 | 95.4 | 52.9 | 67.6 | 59.5 | 77.1 | 96.9 | 92.5 | **98.3**$_{\pm 0.1}$ |
| walker2d-medium-v2 | 59.1 | 52.0 | 72.5 | 79.9 | 77.8 | 75.3 | 74.0 | 73.8 | 76.5 | 79.1 | 79.2 | **81.6**$_{\pm 1.7}$ |
| halfcheetah-medium-expert-v2 | 53.4 | 69.6 | 91.6 | 86.7 | 53.3 | 55.2 | 86.8 | 93.7 | 93.2 | 93.6 | 93.0 | **93.9**$_{\pm 0.1}$ |
| hopper-medium-expert-v2 | 96.3 | 109.1 | 105.4 | 91.5 | 108.7 | 52.5 | 107.6 | 111.1 | 111.1 | 107.6 | 110.4 | **111.5**$_{\pm 3.8}$ |
| walker2d-medium-expert-v2 | 40.1 | 67.3 | 108.8 | 109.6 | 95.6 | 107.5 | 108.1 | 109.0 | 107.7 | 109.3 | 109.6 | **110.4**$_{\pm 0.9}$ |
| Average | 48.2 | 53.8 | 77.6 | 76.5 | 72.9 | 52.6 | 76.2 | 66.2 | 79.1 | 82.4 | 82.2 | **84.1** |
| **Adroit Tasks** | BEAR | BCQ | CQL | IQL | MoRel | EDAC | BC | DT | D-QL | StAR | GDT | VDT |
| pen-human-v1 | -1.0 | 66.9 | 37.5 | 71.5 | -3.2 | 52.1 | 63.9 | 79.5 | 72.8 | 77.9 | 92.5 | **126.7**$_{\pm 4.3}$ |
| hammer-human-v1 | 2.7 | 0.9 | 4.4 | 1.4 | 2.3 | 0.8 | 1.2 | 3.7 | 0.2 | 3.7 | **5.5** | 3.2 $_{\pm 0.3}$ |
| door-human-v1 | 2.2 | -0.05 | 9.9 | 4.3 | 2.3 | 10.7 | 2.0 | 14.8 | 0.0 | 1.5 | 18.6 | **19.7**$_{\pm 0.5}$ |
| pen-cloned-v1 | -0.2 | 50.9 | 39.2 | 37.3 | -0.2 | 68.2 | 37.0 | 75.8 | 57.3 | 33.1 | 86.2 | **145.6**$_{\pm 4.0}$ |
| hammer-cloned-v1 | 2.3 | 0.4 | 2.1 | 2.1 | 2.3 | 0.3 | 0.6 | 3.0 | 3.1 | 0.3 | 8.9 | **19.6**$_{\pm 1.6}$ |
| door-cloned-v1 | 2.3 | 0.01 | 0.4 | 1.6 | 2.3 | 9.6 | 0.0 | 16.3 | 0.0 | 0.0 | 19.8 | **30.6**$_{\pm 0.7}$ |
| Average | 1.0 | 19.8 | 15.6 | 19.7 | 1.0 | 23.6 | 17.5 | 32.2 | 22.2 | 19.4 | 38.9 | **57.6** |
| **Kitchen Tasks** | BEAR | BCQ | CQL | IQL | O-RL | BC | DT | DD | StAR | GDT | DC | VDT |
| kitchen-complete-v0 | 0.0 | 8.1 | 43.8 | 62.5 | 2.0 | 65.0 | 50.8 | 65.0 | 40.8 | 43.8 | 40.9 | **65.9**$_{\pm 0.2}$ |
| kitchen-partial-v0 | 13.1 | 18.9 | 49.8 | 46.3 | 35.5 | 33.8 | 57.9 | 57.0 | 12.3 | 73.3 | 66.8 | **76.1**$_{\pm 10.8}$ |
| Average | 6.6 | 13.5 | 46.8 | 54.4 | 18.8 | 51.5 | 54.4 | 61.0 | 26.6 | 58.6 | 58.7 | **71.0** |
| **Maze2D Tasks** | BEAR | BCQ | CQL | IQL | COMBO | BC | MPPI | DT | QDT | GDT | DC | VDT |
| maze2d-umaze-v1 | 65.7 | 49.1 | 86.7 | 42.1 | 76.4 | 85.7 | 33.2 | 31.0 | 57.3 | 50.4 | 20.1 | **88.0**$_{\pm 4.6}$ |
| maze2d-medium-v1 | 25.0 | 17.1 | 41.8 | 34.9 | 38.5 | 38.3 | 10.2 | 8.2 | 13.3 | 7.8 | 38.2 | **60.3**$_{\pm 0.5}$ |
| Average | 45.35 | 33.1 | 64.3 | 38.5 | 72.5 | 63.6 | 21.7 | 19.6 | 35.3 | 29.1 | 57.6 | **74.2** |
| **AntMaze Tasks** | BEAR | BCQ | CQL | IQL | O-RL | BC | DT | RvS | StAR | GDT | DC | VDT |
| antmaze-umaze-v0 | 73.0 | 78.9 | 74.0 | 87.1 | 64.3 | 54.6 | 59.2 | 65.4 | 51.3 | 76.0 | 85.0 | **100.0**$_{\pm 5.5}$ |
| antmaze-umaze-diverse-v0 | 61.0 | 55.0 | 84.0 | 64.4 | 60.7 | 45.6 | 66.2 | 60.9 | 45.6 | 69.0 | 78.5 | **100.0**$_{\pm 4.7}$ |
| antmaze-medium-diverse-v0 | 8.0 | 0.0 | 53.7 | **70.0** | 0.0 | 0.0 | 7.5 | 67.3 | 0.0 | 0.0 | 0.0 | 30.0$_{\pm 2.8}$ |
| Average | 47.3 | 44.6 | 70.6 | 73.8 | 41.7 | 33.4 | 44.3 | 75.0 | 32.3 | 48.3 | 54.5 | **76.7** |

**Offline Training Performance.** VDT consistently achieves or approaches state-of-the-art performance across all datasets in the pure offline setting, demonstrating the effectiveness of our architecture. The Gym and Adroit environments are characterized by a limited scope of human demonstrations, which leads to extrapolation errors that particularly challenge offline RL. This is precisely why VDT's excellent performance across all tasks can be attributed to its high expressiveness and more effective value guidance. The results of Kitchen tasks requiring generalization to unseen states and long-term

Table 2: Offline-to-online performance of each method, with average rewards reported before (left of arrow) and after (right of arrow) online tuning.

| Dataset | TD3+BC | AWAC | CQL | IQL | PDT | ODT | VDT |
|---|---|---|---|---|---|---|---|
| halfcheetah-medium-replay-v2 | 44.6 → 48.1 | 24.3 → 39.0 | 45.5 → 44.3 | 44.1 → 44.0 | 31.4 → 42.8 | 39.9 → 40.4 | 39.4 → **49.2** |
| hopper-medium-replay-v2 | 60.9 → 90.7 | 77.3 → 79.6 | 95.0 → 95.3 | 92.1 → 93.5 | 84.5 → 94.8 | 86.6 → 88.9 | 96.0 → **119.2** |
| walker2d-medium-replay-v2 | 81.8 → 82.0 | 63.8 → 44.0 | 77.2 → 78.0 | 73.7 → 60.9 | 54.5 → 79.0 | 68.9 → 76.9 | 82.3 → **95.5** |
| halfcheetah-medium-v2 | 48.3 → 50.9 | 37.4 → 41.1 | 44.0 → 29.1 | 47.4 → 48.0 | 39.4 → **69.5** | 42.7 → 42.2 | 43.9 → 53.5 |
| hopper-medium-v2 | 59.3 → 64.6 | 72.0 → 91.0 | 58.5 → 95.7 | 63.8 → 44.3 | 74.4 → 100.2 | 66.9 → 97.5 | 98.3 → **108.1** |
| walker2d-medium-v2 | 83.7 → 85.2 | 30.1 → 79.1 | 72.5 → 89.4 | 79.9 → 68.9 | 63.4 → 88.1 | 72.2 → 76.8 | 81.6 → **89.8** |
| halfcheetah-medium-expert-v2 | 90.7 → 92.1 | 36.8 → 41.0 | 91.6 → 99.9 | 86.7 → 95.3 | 82.6 → 93.3 | 36.8 → 100.9 | 93.9 → **101.7** |
| hopper-medium-expert-v2 | 98.0 → 110.2 | 80.9 → 111.9 | 105.4 → 106.3 | 91.5 → 92.9 | 77.0 → 80.0 | 74.3 → 99.1 | 111.5 → **117.8** |
| walker2d-medium-expert-v2 | 110.1 → 110.1 | 42.7 → 78.3 | 108.8 → 110.1 | 109.6 → 109.6 | 99.1 → 108.9 | 62.0 → 78.7 | 110.4 → **112.7** |
| antmaze-umaze-v0 | 78.6 → 79.1 | 56.7 → 59.0 | 70.1 → 99.4 | 86.7 → 96.0 | 48.6 → 66.8 | 53.1 → 88.5 | 100.0 → **110.0** |
| antmaze-umaze-diverse-v0 | 71.4 → 78.1 | 49.3 → 49.0 | 31.1 → 99.4 | 75.0 → 84.0 | 72.7 → 79.3 | 50.2 → 56.0 | 100.0 → **100.0** |
| antmaze-medium-diverse-v0 | 0.0 → 56.7 | 0.7 → 0.3 | 23.0 → 32.3 | 68.3 → 72.0 | 8.0 → 63.4 | 0.8 → 55.6 | 20.0 → **75.0** |
| Average | 79.0 | 59.4 | 81.6 | 75.78 | 80.51 | 75.13 | **94.38** |

Table 3: Ablation study on model components during offline training. We have abbreviated some task names for simplicity, which does not affect understanding. All experiments are repeated three times, and the average value is taken.

| Advantage Weighting | Regularization | Sampling | hopper-m | walker-m-e | pen-cloned | maze2d-m | antmaze-u |
|---|---|---|---|---|---|---|---|
| ✓ | | | 90.3 | 99.9 | 86.1 | 12.1 | 75.1 |
| | ✓ | | 88.9 | 78.1 | 99.3 | 30.5 | 60.9 |
| | | ✓ | 78.6 | 80.3 | 82.0 | 19.3 | 0.0 |
| ✓ | ✓ | | 95.6 | 103.6 | 131.8 | 40.5 | 95.9 |
| ✓ | ✓ | ✓ | **98.3** | **110.4** | **145.6** | **60.3** | **100.0** |

value optimization demonstrate that VDT can learn useful data features from offline trajectories, enhancing generalization and stability. For the Maze2d environment, which serves as a benchmark to evaluate the capacity of algorithms to stitch segments of disparate trajectories effectively, the performance of VDT significantly outperforms other methods, demonstrating the advantage of the value functions in stitching high-quality trajectories. The AntMaze environment is characterized by sparse rewards and many suboptimal trajectories, which present an even more significant challenge. The performance results of VDT demonstrate the effectiveness and generalizability of the architecture we designed, particularly in antmaze-umaze-diverse tasks.

**Online Tuning Performance.** We conduct online tuning of VDT and observe from the results that VDT achieves optimal or competitive performance across nearly all tasks (as shown in Table 2). These results demonstrate that value-guided methods retain their advantage even in online settings. We attribute the strong performance of value-guided methods in the online setting to their ability to rapidly extract and leverage the Markovian structure of the environment through interaction. This inductive bias allows value-based approaches to adapt efficiently with limited data. In contrast, CSM lacks explicit mechanisms for modeling state transitions and thus may struggle in settings that require fast generalization from sparse interactions.

Compared to methods like ODT, which are restricted to a single setting, VDT achieves optimal performance in offline and offline-to-online scenarios. This powerfully demonstrates the inherent potential of value-guided strategies.

## 5.2 Ablation Study

**Role of Different Components.** As shown in Table 3, we conduct an ablation study on the three key components involving the value guidance in VDT. The results show that using either advantage weighting or individual regularization can improve baseline performance. However, the effectiveness varies significantly across tasks. When both components are applied together (as shown in the fourth row), performance improves substantially, suggesting that advantage weighting and regularization complement policy learning. Specifically, advantage weighting is an important form of sampling

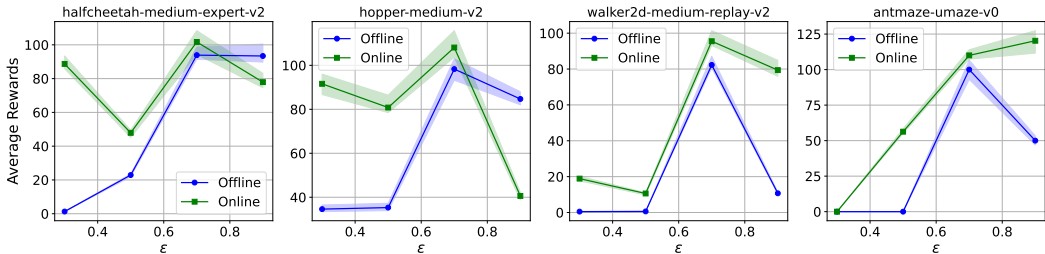

Figure 2: Ablation on the hyperparameter $\epsilon$.

that encourages the policy to favor high-return behaviors. At the same time, the regularization term penalizes low-value actions, effectively constraining the policy within high-value regions. Furthermore, incorporating Q-value-guided sampling enhances the stability and robustness of policy execution. When all three components are combined (as in the last row), the model achieves the best overall performance and consistency across tasks.

**Computational complexity.** Table 4 compares IQL, ODT, and VDT regarding memory usage, parameter count, and training time during offline training and online tuning. Although IQL has the lowest compu-

Table 4: Ablation on the computational complexity.

| Complexity | Offline Training | | | Online Tuning | | |
|---|---|---|---|---|---|---|
| | IQL | ODT | VDT | IQL | ODT | VDT |
| Memory ↓ | 960 M | 3968 M | 4024 M | 960 M | 3968 M | 4024 M |
| Params ↑ | 0.60 M | 5.01 M | 5.24 M | 0.60 M | 5.01 M | 5.24 M |
| Clock Time ↓ | ≈ 1.0 h | ≈ 9.0 h | ≈ 5.0 h | ≈ 1.0 h | ≈ 4.5 h | ≈ 4.0 h |

tational cost, it inevitably suffers from performance limitations. VDT has a slightly higher parameter count than ODT due to the introduction of the value function. However, the value-guided training process leads to faster convergence and better performance in both offline and online stages. Therefore, the slight increase in parameters is considered acceptable. VDT balances model capacity and computational efficiency well, maintaining strong representational power while significantly reducing time costs.

**Impact of evaluation horizon** $E$**.** We investigated the impact of the evaluation horizon during the sampling process (shown in Figure 3). Specifically, we analyzed how varying the evaluation horizon under the guidance of the value function affects model performance. It is well known that a short evaluation horizon may lead to myopic policies, while an overly long horizon can significantly slow down evaluation. Ablation studies show that a horizon length of 5 yields the best performance. Further increasing the horizon results in varying performance trends depending on the task, which is related to the dataset's quality and the rewards' sparsity. For simplicity, we set the evaluation horizon to 5 for all tasks.

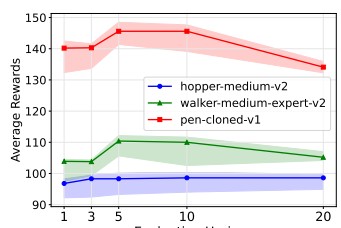

Figure 3: Ablation: Effect of Evaluation Horizon $E$ Length on Offline Training Performance of VDT.

**Impact of context lengths** $K$**.** As shown in Table 5, we observed that VDT performance improved to varying degrees with increasing context length, indicating that VDT exhibits excellent extendability. When $K = 60$, VDT achieved highest average performance. However, increasing context length unquestionably would significantly increase model com-

Table 5: Offline training performance of VDT with different context lengths ($K$) on Gym tasks.

| Datasets | VDT (8) | VDT (20) | VDT (60) | VDT(120) |
|---|---|---|---|---|
| halfcheetah-medium | 28.6 | 43.9 | **44.6** | 43.0 |
| hopper-medium | 77.0 | 98.3 | **99.1** | 65.4 |
| walker2d-medium | 52.6 | **81.6** | 79.9 | 80.5 |
| halfcheetah-medium-expert | 89.5 | **93.9** | **93.9** | 77.0 |
| hopper-medium-expert | 109.3 | 111.5 | **112.7** | 111.2 |
| walker2d-medium-expert | 100.6 | **110.4** | **110.4** | 103.8 |
| **Average** | 76.3 | 89.9 | **90.1** | 80.2 |

plexity and computational cost. Considering the trade-off between performance, complexity, and ensuring fair comparison, we set $K = 20$.

**Impact of hyperparameter $\epsilon$.** In expectile regression, the parameter $\epsilon$ controls the value function's preference over different TD targets. Such preference plays a crucial role in sparse reward settings or high-variance environments, as it helps prevent the policy from being misled by "lucky" samples. As shown in Figure 2, setting $\epsilon = 0.7$ is generally effective across most tasks in both offline and online scenarios. When $\epsilon$ approaches larger values, performance improvements are observed only on the AntMaze tasks, which we attribute to their reliance on trajectory stitching.

**Comparison with stitching baselines.** We conducted a further comparison between VDT and EDT [53], which specializes in trajectory stitching. The core idea behind EDT's ability to stitch trajectories lies in adaptive history truncation. During inference, EDT dynamically selects the optimal history length to maximize the maximum achievable return from the current state. EDT does not use a Q-function to guide action selection. Instead, it forgets unsuccessful histories, which allows the model to escape from low-return trajectories and connect to more optimal trajectory branches.

In addition, we have included a comparison with QDT [54], which combines the dynamic programming capabilities of Q-learning with the sequence modeling strengths of DT, enabling relabeling of return targets and thus improving DT's ability to stitch suboptimal trajectories. VDT differs in that it explicitly introduces a Q-function to provide value-guided weighting for action selection. At the same time, it employs multi-step Bellman equations and double Q-networks to stabilize value function training, thereby extending VDT to be applicable in both offline and online settings, and enabling parallel decision-making during sampling to leverage the evaluation capability of the Q-function fully. As shown in Table 6, VDT demonstrates a clear advantage over EDT and QDT across a wide range of tasks.

Table 6: Performance comparison of EDT, QDT, and VDT on Gym tasks.

| Datasets | EDT | QCS | VDT |
|---|---|---|---|
| halfcheetah-medium | 42.5 | 42.3 | 43.9 |
| hopper-medium | 63.5 | 66.5 | 98.3 |
| walker2d-medium | 72.8 | 67.1 | 81.6 |
| halfcheetah-medium-replay | 37.8 | 35.6 | 39.4 |
| hopper-medium-replay | 89.0 | 52.1 | 96.0 |
| walker2d-medium-replay | 74.8 | 58.2 | 82.3 |
| **Average** | 63.4 | 53.6 | 73.6 |

**Performance comparison in Atari environments.** We evaluate the proposed VDT on the image-based Atari dataset, with results averaged over three random seeds (Table 7). VDT achieves the highest average score across all considered games, substantially outperforming the prior methods CQL, DT, and DC. Specifically, VDT sets new benchmarks in the majority of tasks, evidencing its enhanced ability to handle high-dimensional visual observations. These results demonstrate that VDT possesses superior generalization capability and robustness, validating its effectiveness not only on text-based environments but also in challenging visual domains. The consistent improvements highlight the benefit of our model design in leveraging sequential and high-level representations, thereby providing a unified solution for diverse decision-making scenarios.

Table 7: Performance Comparison in Atari Environments.

| Game | CQL | DT | DC | VDT |
|---|---|---|---|---|
| Breakout | 211.1 | 242.4 | 352.7 | **420.8** |
| Qbert | **104.2** | 28.8 | 67.0 | 69.4 |
| Pong | 111.9 | 105.6 | 106.5 | **113.9** |
| Seaquest | 1.7 | **2.7** | 2.6 | 3.9 |
| Frostbite | 9.4 | 25.6 | 27.8 | **28.9** |
| **Average** | 87.7 | 81.0 | 111.3 | **127.4** |

## 6 Conclusion

In this work, we propose the Value-Guided Decision Transformer (VDT), which organically integrates policy improvement with behavior cloning, enabling efficient trajectory stitching and decision-making through components such as advantage-weighted learning and value regularization. Experiments on diverse RL benchmarks demonstrate that VDT achieves competitive performance in offline and online settings, particularly excelling in stochastic scenarios and suboptimal data regimes. This work establishes a unified CSM architecture for generalizable RL, paving the way for scalable and robust transformer-based policies in real-world applications.

**Limitation.** VDT relies on parallel trajectory evaluation during sampling, which introduces some additional computational cost, though this is generally manageable with GPU batching.

## Acknowledgments and Disclosure of Funding

This work is supported by the National Key Research and Development Program of China (2023YFC2705700), the National Natural Science Foundation of China (Grant No. 62225113, U23A20318, 62576364 and 62276195), the Foundation for Innovative Research Groups of Hubei Province (Grant No. 2024AFA017), the Science and Technology Major Project of Hubei Province (Grant No. 2024BAB046), the Shenzhen Basic Research Project (Natural Science Foundation) Basic Research Key Project (NO. JCYJ20241202124430041), the CCF-Tencent Rhino-Bird Open Research Fund (NO. CCF-Tencent RAGR20250114) and Tencent JR2025TEG002. Dr. Tao's research is partially supported by NTU RSR and Start Up Grants. The numerical calculations in this paper have been done on the supercomputing system in the Supercomputing Center of Wuhan University.

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

# Appendix

The appendix is organized into several sections, each providing additional insights and details related to different aspects of the main work.

---

# A  Proof of Theroem 4.1

Let $\pi_{DT}^*$ be the optimal policy obtained by solving Equation 5. For any $s \in S$, we have:

1. **Support Constraint:** $\pi_{DT}^*(s) \in \mathrm{supp}(\beta(\cdot|s))$,
2. **Value Improvement:** $V^{\pi_{DT}^*}(s) \geq V^\beta(s)$,

where $\mathrm{supp}(\beta(\cdot|s)) = \{a \in \mathcal{A} \mid \exists(s,a) \in \mathcal{D}\}$, and $\pi^*(\mathbf{a} \mid \mathbf{s}) = 0$ whenever $\beta(\mathbf{a} \mid \mathbf{s}) = 0$.

**Lemma A.1** (Implicit Support Constraint). *The policy gradient update satisfies:*

$$\nabla\mathcal{L}(\pi_{DT}^{(d)}) = \mathbb{E}_{\mathcal{D}}\left[2e^{\eta A_t}(\pi_{DT}^{(d)}(\tau_t) - a_t) - \lambda\nabla_a Q_{\hat{\theta}}(s_t, a)\big|_{a=\pi_{DT}^{(d)}(\tau_t)}\right], \tag{7}$$

*where all gradient components vanish outside* $\mathrm{supp}(\beta(\cdot|s))$.

**Lemma A.2** (Policy Improvement). *The advantage-weighted value function [36] satisfies:*

$$\tilde{V}^{\pi_{DT}^{(d+1)}}(s) \geq \tilde{V}^{\pi_{DT}^{(d)}}(s) + \frac{\lambda}{1-\gamma}\mathbb{E}_{\mathcal{D}}\left[\min_i Q_{\hat{\theta}_i}(s, \pi_{DT}^{(d+1)}(s))\right]. \tag{8}$$

*Proof.* Define the policy optimization sequence $\{\pi_{DT}^{(d)}\}_{d=0}^{\infty}$ where $d$ denotes training iterations, with $\pi_{DT}^{(0)} = \beta$. We prove each statement in turn.

**(1) Support Constraint:** From the gradient expression in Lemma A.1, the policy update rule is given by:

$$\nabla\mathcal{L}(\pi_{DT}^{(d)}) = \mathbb{E}_{\mathcal{D}}\left[2e^{\eta A_t}(\pi_{DT}^{(d)}(\tau_t) - a_t) - \lambda\nabla_a Q_{\hat{\theta}}(s_t, a)\big|_{a=\pi_{DT}^{(d)}(\tau_t)}\right], \tag{9}$$

where all terms are evaluated only on trajectories $(s_t, a_t)$ from the dataset $\mathcal{D}$. The gradient thus vanishes outside $\mathrm{supp}(\beta(\cdot|s))$.

Moreover, the behavior cloning component implicitly restricts $\pi_{DT}^{(d)}(s)$ to lie within the convex hull of actions observed in the dataset:

$$\pi_{DT}^*(s) = \lim_{d\to\infty} \mathrm{Proj}_{\mathcal{A}_{\mathcal{D}}(s)}\left(\pi_{DT}^{(d)}(s)\right), \tag{10}$$

where $\mathcal{A}_{\mathcal{D}}(s) := \{a \mid (s,a) \in \mathcal{D}\}$. In addition, the Q-regularization term enforces vanishing gradients near the boundary of this set, preventing the policy from drifting outside. Hence, the optimal policy satisfies the support constraint.

**(2) Value Improvement:** Define the value gap at iteration $d$ as:

$$\epsilon^{(d)} := V^*(s) - \tilde{V}^{\pi_{DT}^{(d)}}(s), \tag{11}$$

where $\tilde{V}^{\pi}$ denotes the advantage-weighted surrogate value used in the optimization.

From Lemma A.2, the policy update guarantees monotonic improvement:

$$\tilde{V}^{\pi_{DT}^{(d+1)}}(s) \geq \tilde{V}^{\pi_{DT}^{(d)}}(s) + \frac{\lambda}{1-\gamma}\mathbb{E}_{\mathcal{D}}\left[\min_i Q_{\hat{\theta}_i}(s, \pi_{DT}^{(d+1)}(s))\right]. \tag{12}$$

Applying the $n$-step Bellman operator $\mathcal{T}^n$ yields:

$$\epsilon^{(d+1)} \leq \gamma^n \epsilon^{(d)} - \lambda\mathbb{E}_{\mathcal{D}}\left[\min_i Q_{\hat{\theta}_i}(s, \pi_{DT}^{(d+1)}(s))\right] \leq \gamma^n \epsilon^{(d)} - \lambda\left(V^{\pi_{DT}^{(d+1)}}(s) - V^\beta(s)\right), \tag{13}$$

where we used that $\min_i Q_{\hat{\theta}_i}$ lower bounds the true $V^{\pi_{DT}^{(d+1)}}$ and that $V^\beta$ is the value under the behavior policy.

Telescoping this recurrence and assuming convergence as $d \to \infty$, we obtain:

$$V^{\pi_{DT}^*}(s) \geq V^\beta(s). \tag{14}$$

The inequality is strict unless the Q-functions are constant over $\mathcal{A}_{\mathcal{D}}(s)$, in which case the behavior policy $\beta$ is already optimal within the dataset support.

# B    Proof of Theroem 4.2

Let $\pi^*$ denote the globally optimal policy and $\pi_{DT}^*$ denote the optimal policy constrained to dataset support. Define $\mathcal{A}_D(s) := \{a \mid (s, a) \in \mathcal{D}\}$ to be the set of actions observed in the dataset at state $s$. For any $s$, we have:

$$V^{\pi^*}(\mu) - V^{\pi_{DT}^*}(\mu) \leq \frac{2\gamma}{(1-\gamma)^2}\mathbb{E}_{s\sim d^{\pi^*}}\left[\max_{a\notin\mathcal{A}_D(s)} Q^{\pi^*}(s,a) - \max_{a\in\text{supp}(\beta(\cdot|s))} Q^{\pi^*}(s,a)\right] \tag{15}$$

*Proof.* **Step 1: Performance Difference Lemma**

Recall for any two policies $\pi, \pi'$, the performance difference lemma states:

$$V^{\pi}(\mu) - V^{\pi'}(\mu) = \frac{1}{1-\gamma}\mathbb{E}_{s\sim d^\pi}\left[\mathbb{E}_{a\sim\pi(\cdot|s)}\left[Q^{\pi'}(s,a) - V^{\pi'}(s)\right]\right] \tag{16}$$

where $d^\pi(s)$ is the normalized discounted state distribution under $\pi$.

**Step 2: Comparing $\pi^*$ and $\pi_{DT}^*$**

Applying the lemma yields:

$$V^{\pi^*}(\mu) - V^{\pi_{DT}^*}(\mu) = \frac{1}{1-\gamma}\mathbb{E}_{s\sim d^{\pi^*}}\left[\mathbb{E}_{a\sim\pi^*(\cdot|s)}\left[Q^{\pi_{DT}^*}(s,a) - V^{\pi_{DT}^*}(s)\right]\right] \tag{17}$$

$$\leq \frac{1}{1-\gamma}\mathbb{E}_{s\sim d^{\pi^*}}\left[\max_a Q^{\pi_{DT}^*}(s,a) - \max_{a\in\mathcal{A}_D(s)} Q^{\pi_{DT}^*}(s,a)\right] \tag{18}$$

since $\pi^*$ may select actions $a^* \notin \mathcal{A}_D(s)$ that $\pi_{DT}^*$ cannot, incurring a value gap.

**Step 3: Relating $Q^{\pi_{DT}^*}$ to $Q^{\pi^*}$**

Since $Q^{\pi_{DT}^*}(s,a) \leq Q^{\pi^*}(s,a)$ for all $a$, we may further bound:

$$V^{\pi_{DT}^*}(s) = \max_{a\in\mathcal{A}_D(s)} Q^{\pi_{DT}^*}(s,a) \leq \max_{a\in\mathcal{A}_D(s)} Q^{\pi^*}(s,a) \tag{19}$$

and

$$V^{\pi^*}(s) = \max_a Q^{\pi^*}(s,a) \tag{20}$$

At each state $s$, the maximal loss incurred is thus

$$\delta(s) := V^{\pi^*}(s) - \max_{a \in \mathcal{A}_D(s)} Q^{\pi^*}(s, a) = \max_a Q^{\pi^*}(s, a) - \max_{a \in \mathcal{A}_D(s)} Q^{\pi^*}(s, a) \tag{21}$$

**Step 4: Cumulative suboptimality by recursion**

Because $\pi_{DT}^*$ can only select actions within the dataset support at each step, the loss $\delta(s_t)$ compounds over all timesteps. By recursive expansion and geometric series, we obtain:

$$V^{\pi^*}(\mu) - V^{\pi_{DT}^*}(\mu) \leq \frac{1}{1-\gamma}\mathbb{E}_{s \sim d^{\pi^*}}[\delta(s)] + \frac{\gamma}{1-\gamma}\mathbb{E}_{s \sim d^{\pi^*}}[\delta(s)] + \cdots \tag{22}$$

Summing the geometric series yields:

$$V^{\pi^*}(\mu) - V^{\pi_{DT}^*}(\mu) \leq \frac{1}{(1-\gamma)^2}\mathbb{E}_{s \sim d^{\pi^*}}[\delta(s)] \tag{23}$$

By a refinement [41], accounting for effect propagation across steps, a factor $2\gamma$ appears:

$$V^{\pi^*}(\mu) - V^{\pi_{DT}^*}(\mu) \leq \frac{2\gamma}{(1-\gamma)^2}\mathbb{E}_{s \sim d^{\pi^*}}[\delta(s)] \tag{24}$$

Thus, we have proven

$$V^{\pi^*}(\mu) - V^{\pi_{DT}^*}(\mu) \leq \frac{2\gamma}{(1-\gamma)^2}\mathbb{E}_{s \sim d^{\pi^*}}\left[\max_{a \notin \mathcal{A}_D(s)} Q^{\pi^*}(s, a) - \max_{a \in \mathrm{supp}(\beta(\cdot|s))} Q^{\pi^*}(s, a)\right] \tag{25}$$

which bounds the loss in optimal value due to dataset support constraints.

# C  Hyperparameters Configuration

Table 8: Hyperparameter configurations for offline training and online tuning.

| Hyperparameter | Offline Training | Online Tuning |
|---|---|---|
| Context Length $K$ | 20 | 20 |
| Batch Size | 512 | 512 |
| Training Steps | 10000 | 25000 |
| Learning Rate | 3e-4 | 1e-4 |
| Weight Decay | 1e-4 | — |
| Number of Layers | 6 | 6 |
| Attention Heads | 4 | 4 |
| Embedding Dimension | 256 | 256 |
| Activation | GeLU | GeLU |
| Dropout | 0.1 | 0.1 |
| Discount $\gamma$ | 0.99 | 0.99 |
| Threshold $\epsilon$ | 0.7 | 0.7 |
| Inverse Temperature $\eta$ | 3 | 3 |
| Balance Coefficient $\lambda$ | 0.5 | 0.5 |
| pct_traj | 1 | 1 |
| Updates between Rollouts | — | 300 |
| Gradient Norm Clip | 0.25 | 0.25 |
| Replay Buffer Size | — | 1000 |
| Q-network Layers | 2 | 2 |
| Q-network Width | 256 | 256 |

# D  Algorithm Pseudocode

---

**Algorithm 1** Offline Training with VDT

---

**Input:** Offline dataset $\mathcal{D}_{\text{offline}}$, Context length $K$, Expectile $\epsilon$, Discount $\gamma, \lambda, \eta$
**Initialize:** $\pi_{DT}, V_\psi, Q_{\theta_1}, Q_{\theta_2}, Q_{\hat{\theta}_1}, Q_{\hat{\theta}_2}$
**for** iteration $= 1$ to $T$ **do**
    Sample batch $\mathcal{B} \sim \mathcal{D}_{\text{offline}}$
    {Value network update}
    Update $V_\psi$ using: $\mathbb{E}_{(s,a)\sim\mathcal{B}}[L_2^\epsilon(Q_{\hat{\theta}}(s,a) - V_\psi(s))]$
    {Q-network update}
    Compute n-step targets: $y = \sum_{k=0}^{n-1} \gamma^k r_k + \gamma^n V_\psi(s_{t+n})$
    Update $Q_{\theta_i}$ using: $\mathbb{E}_\mathcal{B}[(y - Q_{\theta_i}(s_t, a_t))^2]$ for $i = 1, 2$
    {Advantage-weighted policy update}
    Compute advantages: $A_t = \min_i Q_{\hat{\theta}_i}(s_t, a_t) - V_\psi(s_t)$
    Update $\pi_{DT}$ using:
        $\mathbb{E}_\mathcal{B}[\exp(\eta A_t)\|\pi_{DT}(\tau_t) - a_t\|^2 - \lambda \min_i Q_{\hat{\theta}_i}(s_t, \pi_{DT}(\tau_t))]$
    {Target network update}
    $\hat{\theta}_i \leftarrow \rho\hat{\theta}_i + (1 - \rho)\theta_i$ for $i = 1, 2$
**end for**

---

**Algorithm 2** Online Tuning with Trajectory Replay

---

**Input:** Pretrained $\pi_{DT}, Q_{\hat{\theta}_i}$, Online RTG $g_{\text{online}}$, Buffer size $N$, Context length $K$
Initialize replay buffer $\mathcal{T}_{\text{replay}} \leftarrow$ Top-$N(\mathcal{D}_{\text{offline}})$
**for** round $= 1$ to $R$ **do**
    Generate trajectory $\tau$ with $\pi_{DT}$ using $g_{\text{online}}$
    Relabel RTG: $g_t = \sum_{j=t}^{|\tau|} r_j$ for $t \in \tau$
    Update $\mathcal{T}_{\text{replay}}$ (FIFO)
    **for** gradient step $= 1$ to $I$ **do**
        Sample trajectories $\{\tau_j\} \sim \mathcal{T}_{\text{replay}}$ with $p(\tau) \propto |\tau|$
        **for** each $\tau_j$ **do**
            Sample sub-trajectory $(\hat{s}_{t:t+K}, \hat{a}_{t:t+K}, \hat{g}_{t:t+K})$
            Compute $A_t = \min_i Q_{\hat{\theta}_i}(\hat{s}_t, \hat{a}_t) - V_\psi(\hat{s}_t)$
        **end for**
        Update $\pi_{DT}$ using:
            $\frac{1}{B} \sum \left[\exp(\eta A_t)\|\pi_{DT}(\hat{s}_t, \hat{g}_t) - \hat{a}_t\|^2 - \lambda \min_i Q_{\hat{\theta}_i}(\hat{s}_t, \pi_{DT}(\hat{s}_t, \hat{g}_t))\right]$
        Update $Q_{\theta_i}$ using online transitions
        $\hat{\theta}_i \leftarrow \rho\hat{\theta}_i + (1 - \rho)\theta_i$
    **end for**
**end for**

---

## E  Environment Details

**Gym tasks**: The Gym-MuJoCo tasks (hopper, halfcheetah, walker2d) are popular benchmarks used in offline deep RL. They are relatively straightforward and characterized by datasets with a significant proportion of near-optimal trajectories and smooth reward functions. The "medium" dataset is generated by first training a policy online using Soft Actor-Critic, early-stopping the training, and collecting 1M samples from this partially-trained policy. The "random" datasets are generated by unrolling a randomly initialized policy on these three domains. The "medium-replay" dataset consists of recording all samples in the replay buffer observed during training until the policy reaches the "medium" level of performance. Datasets similar to these three have been used in prior work, but in order to evaluate algorithms on mixtures of policies, we further introduce a "medium-expert" dataset by mixing equal amounts of expert demonstrations and suboptimal data, generated via a partially trained policy or by unrolling a uniform-at-random policy.

**Adroit tasks**: The Adroit domain involves controlling a 24-DoF simulated Shadow Hand robot to perform tasks such as hammering a nail, opening a door, twirling a pen, or picking up and moving a ball. This domain is chosen to study the impact of narrow expert data distributions and human

---

**Algorithm 3** Sampling Process

---

**Input:** Initial state $s_0$, candidate RTGs $\{\hat{r}_0^1, ..., \hat{r}_0^m\}$, Evaluation horizon $E$, Discount $\gamma$, Policy $\pi_{DT}$, Q-networks $Q_{\hat{\theta}_1}, Q_{\hat{\theta}_2}$

**Initialize:** Current state $s_t \leftarrow s_0$, Active trajectories $\{\tau^k\}_{k=1}^m \leftarrow \{(s_0, \hat{r}_0^k)\}_{k=1}^m$, Target Q-networks $Q_{\hat{\theta}_i}$

**while** not termination condition **do**
    *// Parallel candidate action generation*
    **for** $k = 1$ **to** $m$ **in parallel do**
        Sample action $a_t^k \sim \pi_{DT}(\tau^k)$
    **end for**
    *// Batched trajectory prediction*
    **for** $k = 1$ **to** $m$ **in parallel do**
        Initialize predicted trajectory $\tau_{\text{pred}}^k \leftarrow (s_t, a_t^k)$
        Initialize cumulative Q-value $Q_{\text{total}}^k \leftarrow 0$
        **for** $i = 0$ **to** $E - 1$ **do**
            Predict next state: $s_{t+i+1}^k \leftarrow \text{EnvModel}(\tau_{\text{pred}}^k)$
            Sample next action: $a_{t+i+1}^k \sim \pi_{DT}(\tau_{\text{pred}}^k)$
            Compute Q-value: $q_i^k = \min_{j=1,2} Q_{\hat{\theta}_j}(s_{t+i}^k, a_{t+i}^k)$
            Accumulate: $Q_{\text{total}}^k \leftarrow Q_{\text{total}}^k + \gamma^i q_i^k$
            Append $(a_{t+i+1}^k, s_{t+i+1}^k)$ to $\tau_{\text{pred}}^k$
        **end for**
    **end for**
    *// Optimal action selection*
    Select optimal index: $k^* \leftarrow \arg\max_{1 \leq k \leq m} Q_{\text{total}}^k$
    Execute action: $a_t \leftarrow a_t^{k^*}$
    *// Environment interaction & trajectory update*
    Observe reward $r_t$, next state $s_{t+1}$ from environment
    **for** $k = 1$ **to** $m$ **do**
        Update RTG: $\hat{r}_{t+1}^k \leftarrow \hat{r}_t^k - r_t$
        Append transition: $\tau^k \leftarrow \tau^k \oplus (a_t, r_t, s_{t+1}, \hat{r}_{t+1}^k)$
    **end for**
    $t \leftarrow t + 1$
**end while**

---

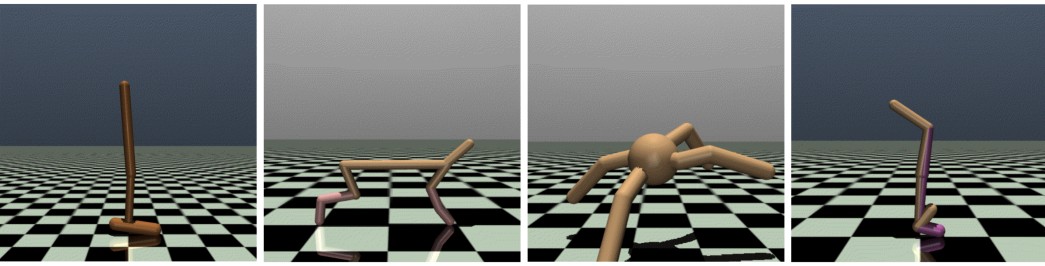

Figure 4: Illustration of Gym environments [39].

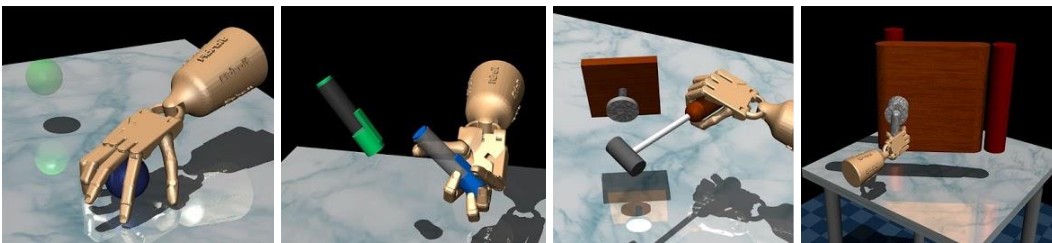

Figure 5: Illustration of Adroit environments [39].

demonstrations on sparse-reward, high-dimensional robotic manipulation tasks. Since these tasks are primarily derived from human behavior, they exhibit a limited state-action space, requiring robust policy regularization to ensure consistent agent performance. The Adroit domain has several unique properties that make it qualitatively different from the Gym tasks. First, the data is collected from human demonstrators. Second, each task is difficult to solve with online RL due to sparse rewards and exploration challenges, which make cloning and online RL alone insufficient. Lastly, the tasks are high-dimensional, presenting a representation learning challenge.

**Kitchen tasks**: The Kitchen domain involves controlling a 9-DoF Franka robot in a kitchen environment with everyday household items such as a microwave, kettle, overhead light, cabinets, and an oven. The goal is to interact with these items to achieve a desired state configuration. This domain benchmarks the impact of multitasking behaviour in a realistic, non-navigation environment, where the "stitching" challenge arises from complex paths through the state space. Consequently, algorithms must generalize to unseen states rather than rely solely on training trajectories. The environment requires the agent to complete multiple sequential sub-tasks, further emphasizing the need for robust generalization. The "complete" dataset consists of the robot performing all the desired tasks in order. This

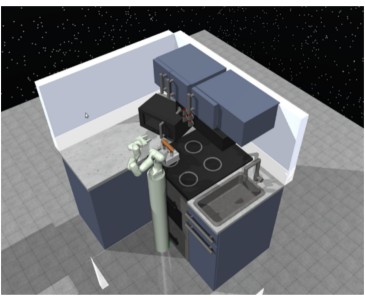

Figure 6: Illustration of Kitchen environments [39].

provides data that is easy for an imitation learning method to solve. The "partial" dataset consists of undirected data, where the robot performs subtasks that are not necessarily related to the goal configuration. In the "partial" dataset, a subset is guaranteed to solve the task, meaning an imitation learning agent may learn by selectively choosing the proper subsets of the data.

**Maze2D tasks**: The Maze2D domain is a navigation task in which a 2D agent must reach a fixed goal location. It tests offline RL algorithms' ability to stitch together previously collected sub-trajectories to find the shortest path to the goal. Three maze layouts are provided: the "maze", "medium", and "large" mazes. These tasks evaluate the algorithm's capability to effectively combine sub-trajectories and identify the shortest path to the set goal. The data is generated by selecting goal locations randomly and then using a planner that generates sequences of waypoints, followed by a PD controller. The trajectories in the dataset are visualized in Appendix G. Because the controllers memorize the reached waypoints, the data collection policy is non-Markovian.

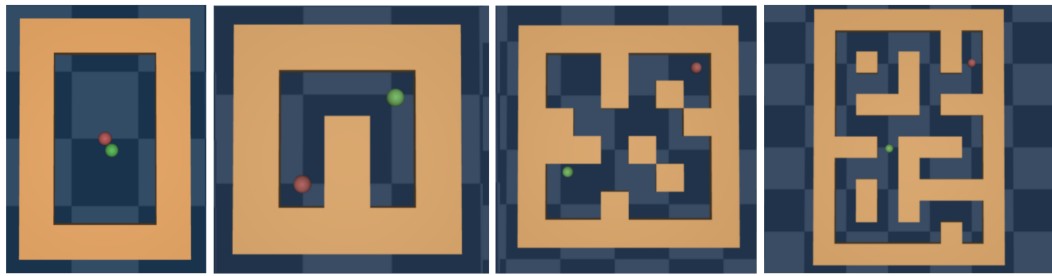

Figure 7: Illustration of Maze2D and AntMaze environments [39].

**AntMaze tasks**: The AntMaze domain extends the Maze2D task by replacing the 2D ball with a more complex 8-DoF "Ant" quadruped robot, presenting a more demanding navigation challenge. This domain is introduced to test the stitching challenge with a morphologically complex robot, better representing real-world robotic navigation tasks. The task uses a sparse 0-1 reward, activated upon reaching the goal. The data is generated by training a goal-reaching policy and using it with the same high-level waypoint generator from maze2d to provide subgoals that guide the agent to the goal. As in Maze2D, the controllers for this task are non-Markovian as they rely on tracking visited waypoints.

## F  More Experiments

**Impact of model size.** We investigate the impact of model size on various behavior cloning and decision transformer variants, including BC, DT, ODT, and VDT, under both offline and online

Table 9: The size and the average and standard deviation of the normalized reward in our experiments.

| Dataset | Size | Normalized Reward |
|---|---|---|
| halfcheetah-medium-replay-v2 | 202000 | $27.17 \pm 15.79$ |
| hopper-medium-replay-v2 | 402000 | $14.98 \pm 16.32$ |
| walker2d-medium-replay-v2 | 302000 | $14.84 \pm 19.48$ |
| halfcheetah-medium-v2 | 1000000 | $40.68 \pm 5.12$ |
| hopper-medium-v2 | 999906 | $44.32 \pm 12.27$ |
| walker2d-medium-v2 | 999995 | $62.09 \pm 23.83$ |
| halfcheetah-expert-v2 | 1000000 | $80.30 \pm 35.82$ |
| hopper-medium-expert-v2 | 999906 | $100.59 \pm 31.66$ |
| walker2d-medium-expert-v2 | 999995 | $112.09 \pm 23.83$ |
| pen-human-v1 | 4800 | $202.69 \pm 154.48$ |
| hammer-human-v1 | 10948 | $23.80 \pm 33.36$ |
| door-human-v1 | 6504 | $28.35 \pm 13.88$ |
| pen-cloned-v1 | 499886 | $108.63 \pm 122.43$ |
| hammer-cloned-v1 | 999872 | $8.11 \pm 23.35$ |
| door-cloned-v1 | 999939 | $12.29 \pm 18.35$ |
| kitchen-complete-v0 | 999800 | $62.25 \pm 19.83$ |
| kitchen-partial-v0 | 999800 | $89.49 \pm 14.15$ |
| maze2d-umaze-v1 | 999869 | $-12.55 \pm 9.82$ |
| maze2d-medium-v1 | 1999733 | $-3.46 \pm 3.95$ |
| antmaze-umaze-v0 | 998573 | $86.14 \pm 34.55$ |
| antmaze-medium-diverse-v0 | 999930 | $6.36 \pm 10.07$ |
| antmaze-umaze-diverse-v0 | 999000 | $3.48 \pm 18.32$ |

settings. As shown in Table 10, performance generally improves with increased model capacity. For example, across all three datasets—halfcheetah, hopper, and walker2d—VDT (online) consistently achieves the best performance for each model size, with scores improving as we move from the smallest configuration (3,1,256) to the largest (12,12,768). The performance saturates or drops slightly on walker2d in the largest model, suggesting that model complexity must be matched with task difficulty and data availability. Similarly, other variants like ODT and DT also benefit from larger models, though the gain is more moderate than VDT. Interestingly, the offline methods also demonstrate strong performance with moderate-size models, especially VDT (offline), which performs competitively or better than its online variant in smaller models. For instance, in the (6,4,256) configuration, VDT (offline) achieves 98.3 on hopper, matching the best result at this size. These results suggest that while increasing model size generally boosts performance, particularly for VDT, the returns diminish and may even reverse if the model becomes too large relative to the dataset, likely due to overfitting or optimisation difficulty in reinforcement learning scenarios.

**Impact of inverse temperature $\eta$.** Table 11 shows the impact of the inverse temperature $\eta$ on policy performance. As $\eta$ increases from 1 to 3, performance consistently improves, indicating that more substantial advantage weighting helps the model focus on high-value trajectories. In this context, $\eta$ amplifies the difference between actions, guiding the policy toward more optimal behavior. However, setting $\eta$ too high (e.g., $\eta = 10$) leads to performance degradation, likely due to overfitting to a small subset of high-advantage samples and reduced generalization. $\eta = 3$ achieves the best average performance offline and online. This trend highlights the role of $\eta$ in controlling the selectivity of the learning process, where a moderate value strikes a good balance between stability and performance.

Table 10: Ablation on the model size. Model size is denoted as $(x, y, z)$ for number of layers, attention heads, and embedding dimension. **Bold** indicates the best result overall, and underline highlights the best among offline methods.

| Model Size | Method | halfcheetah-medium-v2 | hopper-medium-v2 | walker2d-medium-v2 |
|---|---|---|---|---|
| (3,1,256) | BC (offline) | 34.0 | 44.8 | 72.3 |
| | DT (offline) | 42.7 | 60.3 | 70.6 |
| | ODT (offline) | 18.3 | 62.9 | 57.2 |
| | VDT (offline) | 43.6 | 98.3 | 79.0 |
| | ODT (online) | 23.6 | 70.1 | 52.8 |
| | VDT (online) | **49.0** | **105.9** | **89.8** |
| (6,4,256) | BC (offline) | 42.6 | 52.9 | 75.3 |
| | DT (offline) | 42.6 | 67.6 | 74.0 |
| | ODT (offline) | 42.7 | 66.9 | 72.2 |
| | VDT (offline) | 43.9 | 98.3 | 81.6 |
| | ODT (online) | 42.2 | 97.5 | 76.8 |
| | VDT (online) | **53.5** | **108.1** | **89.8** |
| (12,12,768) | BC (offline) | 42.9 | 67.3 | 69.1 |
| | DT (offline) | 39.3 | 74.9 | 75.2 |
| | ODT (offline) | 42.7 | 79.1 | 66.0 |
| | VDT (offline) | 44.0 | 96.0 | 74.1 |
| | ODT (online) | 43.9 | 99.8 | 82.1 |
| | VDT (online) | **54.9** | **108.9** | **84.2** |

Table 11: Ablation on the hyperparameter $\eta$.

| Dataset | Offline Training | | | | Online Tuning | | | |
|---|---|---|---|---|---|---|---|---|
| | $\eta =1$ | $\eta =3$ | $\eta =5$ | $\eta =10$ | $\eta =1$ | $\eta =3$ | $\eta =5$ | $\eta =10$ |
| halfcheetah-medium-expert-v2 | 34.2 | 93.9 | 88.65 | 59.1 | 77.3 | 101.7 | 10.9 | 103.9 |
| hopper-medium-v2 | 18.3 | 98.3 | 97.6 | 70.0 | 29.6 | 108.1 | 93.1 | 100.3 |
| walker2d-medium-replay-v2 | 65.9 | 82.3 | 82.1 | 46.9 | 58.3 | 95.5 | 96.9 | 22.1 |
| antmaze-umaze-v0 | 50.7 | 100.0 | 90.0 | 88.3 | 39.1 | 110.0 | 110.0 | 75.8 |
| Average | 42.3 | 93.6 | 89.1 | 66.1 | 51.1 | 103.8 | 77.7 | 75.5 |

**Impact of RTG alignment.** Table 12 presents the ablation study on RTG alignment during online tuning. Without RTG alignment, VDT performs worse than offline training on certain tasks, indicating that RTG alignment effectively leverages trajectory signals from online interaction by correcting reward guidance. Incorporating RTG alignment consistently improves performance across all tasks. The average score increases from 75.6 to 94.2, indicating a substantial performance gain. In particular, tasks such as halfcheetah-medium-v2 and walker2d-medium-v2 benefit significantly, demonstrating that RTG alignment is especially effective when dealing with suboptimal trajectories.

Table 13: Ablation study on model components during online tuning. All experiments are repeated three times, and the average value is taken.

| Advantage Weighting | Regularization | hopper-medium-v2 | walker2d-medium-expert-v2 | antmaze-umaze-v0 |
|:---:|:---:|:---:|:---:|:---:|
| ✓ | | 104.1 | 110.2 | 90.5 |
| | ✓ | 107.0 | 94.3 | 100.0 |
| ✓ | ✓ | 108.1 | 112.7 | 110.0 |

Table 12: Ablation study on RTG alignment. Online Tuning-w/o refers to VDT during online tuning without RTG alignment.

| Datasets | Offline Training | Online Tuning-w/o | Online Tuning |
|:---:|:---:|:---:|:---:|
| halfcheetah-medium-replay-v2 | 39.4 | 36.0 | 49.2 |
| hopper-medium-replay-v2 | 96.0 | 99.4 | 119.2 |
| walker2d-medium-replay-v2 | 82.3 | 85.6 | 95.5 |
| halfcheetah-medium-v2 | 43.9 | 11.7 | 53.5 |
| hopper-medium-v2 | 98.3 | 99.6 | 108.1 |
| walker2d-medium-v2 | 81.6 | 44.0 | 89.8 |
| halfcheetah-medium-expert-v2 | 93.9 | 80.6 | 101.7 |
| hopper-medium-expert-v2 | 111.5 | 111.0 | 117.8 |
| walker2d-medium-expert-v2 | 110.4 | 112.3 | 112.7 |
| Average | 84.1 | 75.6 | 94.2 |

**Role of different components.** As shown in Table 13, we also conduct an ablation study on advantage weighting and regularization in online fine-tuning for VDT. Since the sampling stage is independent of the training phase, and we have already demonstrated the advantage of value guidance during sampling in Table 3, we do not perform an additional ablation for the sampling process. We observe that the components of VDT exhibit similar effectiveness in the online setting as in the offline one. Using either component individually leads to performance improvement while combining both yields the best results. This demonstrates that value-guided methods are effective in both online and offline scenarios.

# G    Broader Impact

The Value-Guided Decision Transformer advances decision-making automation by improving adaptability across offline and online settings, offering potential benefits in areas like robotics, logistics, and personalized AI assistance. While designed to enhance efficiency and scalability, its adoption invites considerations around balancing human-AI collaboration—such as ensuring human oversight in critical decisions and avoiding over-reliance on automated systems. By prioritizing transparency in value-guided objectives and fostering partnerships between developers and domain experts, VDT's deployment can support human-centric innovation while addressing practical challenges responsibly.

