# OpenReview forum: "Value-Guided Decision Transformer: A Unified Reinforcement Learning Framework for Online and Offline Settings"
_NeurIPS.cc/2025/Conference — NeurIPS 2025 poster_

### Official Review · Reviewer_8M1K · 2025-06-18

**Clarity:** 3
**Significance:** 3
**Originality:** 3
**Rating:** 5
**Confidence:** 4

**Summary:**

This paper proposes Value-Guided Decision Transformer (VDT), which leverages value functions for offline-to-online finetuning for decision transformers. In the offline stage, a double MLP critic network with TD(n) IQL loss is trained, and the decision transformer is trained via advantage-weighted supervised loss with a Q-value regularizer. For the online stage, the training process is almost the same, except that the replay buffer is trajectory-based, and multiple active trajectories with different RTGs are maintained. On a variety of environments, the proposed method outperforms several baselines.

**Questions:**

I have one question: In Tab. 4, the authors compared the efficiency of their method with IQL. I am a bit curious about the settings of IQL here: why is IQL taking comparable amount of parameters to decision transformer-based method and is much slower during online finetuning? My expectation is that IQL should be a lot faster as it does not need to deal with the heavy (compared to MLP similar to your critic) decision transformer architecture.

**Ethical Concerns:**

["NO or VERY MINOR ethics concerns only"]

**Final Justification:**

The authors addressed my concerns with ample experiment results, which exceeds my expectation. Therefore, I am willing to increase my score to 5.

**Limitations:**

Yes (though I would say a limitation is lack of evaluation on image-based environments; I do not feel this to be a ground for rejection but it is a limitation).

**Quality:**

3

**Strengths And Weaknesses:**

**Strengths**

1. The paper is well-written and very easy to follow; the design of value-augmented decision transformer loss and online stage designs are clearly conveyed. It is also very well-rounded in the sense that it provides detailed hyperparameters, discussions on broader impact, environment specifications and pseudocode that helps the understanding of VDT.

2. The idea is intuitive: adding RL gradients for supervised learning loss, and is supported by theoretical analysis (Thm. 4.1). It is also very interesting to see the idea of sampling from different RTGs with parallel trajectories which tries to address a crucial concern of decision transformers that it could be sensitive to different RTGs.

3. The experiment results seems quite solid. Authors compare their method to a large number of baselines on a variety of environments, which I feel sufficient for proving the effectiveness of the proposed method. There are also extensive ablations on computational complexity (Tab. 4), algorithm components (Tab. 3) and hyperparameters such as context length, architecture, $\eta$ and $\epsilon$ (Fig. 3, Tab. 5, Tab. 8, Tab. 9).

**Weaknesses**

1. While the idea of sampling parallel trajectories is very interesting, it limits the proposed method largely within simulation environments. The paper does not seem to maintain a standalone dynamic / world model to predict future states, nor is there any loss term on predicting states or state prediction output by the decision transformer itself. In this case, when you "predict" next state from "EnvModel" (Alg. 3), you are generating a sequence of $E$-step rollout for each of the $m$ candidate RTGs, and always cancel your actions afterwards until the optimal action is selected. This is a major limitation in real-life scenarios; for example, if you cook a steak, there is no way to "uncook" it.

2. It seems that the implementation of "alignment of RTGs" is not explained in Sec. 4.2 where it is proposed. More specifically, how is the RTG token in the last timestep "matching" the reward obtained by the agent? Ideally the RTG in the last step should be $0$, which should not match the reward obtained by the agent. The mechanism seems somehow explained in Sec. 4.3 and its related content in the appendix; it would be better if the authors can make this clearer when it is introduced.

3. There are some literature in online finetuning decision transformers missing. For example, Yan et al. [1] conducts offline-to-online RL with online decision transformer + TD3; the idea of attaching RL gradients from Q-value to the training objective and trajectory-level replay buffer in its implementation are similar to this paper, and thus is more relevant to this paper than ODT. Another related work that uses RL values for offline decision transformer is Wang et al. [2].

4. The proposed method seems to be sensitive with respect to $\epsilon$.

**Minor Weaknesses**

1. On page 25 in the appendix, a space is missing between . and $\eta=3$.

2. in Fig. 1, the subscripts are out of the circle and almost overlap with other parts (e.g. $s_{t-K+1}$ and $a_{t-K+1}$).

**References**

[1] K. Yan et al. Reinforcement Learning Gradients as Vitamin for Online Finetuning Decision Transformers. In NeurIPS, 2024.

[2] Y. Wang et al. Critic-Guided Decision Transformer for Offline Reinforcement Learning. In AAAI, 2024.

---

> ### Author Rebuttal · Authors · 2025-07-31
>
> Thank you very much for your careful review of our work. We'll answer your questions one by one in the following, including some misunderstandings and some essential academic questions worth exploring.
>
> **W1: The paper does not seem to maintain a standalone dynamic model to predict future states. In this case, when you "predict" next state from "EnvModel" (Alg. 3), you are generating a sequence of $E$-step rollout for each of the candidate RTGs, and always cancel your actions afterwards until the optimal action is selected.**
> We must acknowledge that the approach of sampling parallel trajectories is difficult to implement in scenarios where future states are completely unpredictable. In such cases, we can only rely on a "single-shot" sampling strategy, i.e., single-step sampling with Evaluation Horizon = 1. As shown in the table below, we find that VDT still maintains a advantage over other methods, which is intuitive: the advantage-weighting and penalty mechanisms provided by the value function can further enhance the model's decision-making ability, even when multi-step parallel sampling is not possible.
> ||DT|DC|VDT (Evaluation Horizon=1)
> -|-|-|-
> halfcheetah-medium|42.6 |43.0|42.9
> hopper-medium|67.6|92.5|96.8
> walker2d-medium|74.0|79.2|80.0
> halfcheetah-medium-expert|86.8|93.0|93.6
> hopper-medium-expert|107.6|110.4|111.2
> average score|75.7|83.6|84.9
>
> On the other hand, although most tasks have or can be trained with a simulation environment, it is indeed important to consider our sampling methods in extreme scenarios where no explicit simulator is available. One intuitive approach is to directly add an environment prediction head to VDT, enabling the model to perform parallel sampling during inference autonomously. Another approach is to train a world model to provide future states for VDT, which has been explored in many previous works and is relatively feasible [1][2]. In general, training a dynamic model to predict states and rewards is often slightly easier than training a decision model. If we do proceed in this way, it could be argued that, to some extent, we are essentially equipping the current task with a simulation environment based on the existing data. Due to the limited time for rebuttal, we regret that we are unable to explore these two designs now. We will include experiments and discussions on both methods in the appendix of the revised version.
>
> [1] Ha D, Schmidhuber J. Recurrent world models facilitate policy evolution. NeurIPS, 2018.
> [2] Hafner D, Pasukonis J, Ba J, et al. Mastering diverse control tasks through world models. Nature, 2025.
>
> **W2: The implementation of "alignment of RTGs" is not explained. More specifically, how is the RTG token in the last timestep "matching" the reward obtained by the agent? Ideally the RTG in the last step should be 0, which should not match the reward obtained by the agent.**
> In line 5 of Algorithm 2 in Appendix C, we illustrate the alignment of RTGs, where the RTG at time step $t$ for the current trajectory is calculated as: $RTG_t = \sum_{j=t}^{|\tau|} r_j$, with $|\tau|$ denoting the trajectory length. During trajectory sampling, a trajectory may terminate either due to reaching the interaction limit (1000 steps) or by actually achieving the task goal. Therefore, the reward at the final step (i.e., the reward upon trajectory termination) is more accurately the immediate reward returned by the environment at that step, which may or may not be zero. Even if the goal is achieved, the immediate reward is not necessarily zero. Thus, in the main text, we refer to the RTG of the last token as the reward obtained by the model at the current step. For example, in the maze task, the reward for the final step after reaching the goal is +1. We will further clarify the details regarding the alignment of RTGs in the main text to avoid any ambiguity.
>
> **W3: There are some literature in decision transformers missing[1][2].**
> We have already cited and compared CGDT [2] in the experimental section of the main text. CGDT, designed for offline reinforcement learning, also adopts the DT loss as its foundation, but further introduces the value estimation from an asymmetrically pre-trained Critic as a penalty term, encouraging the policy to select optimistic actions that the Critic evaluates as exceeding the target return. TD3+ODT [1], which largely follows the ODT algorithm pipeline, additionally incorporates the RL gradient from the Critic as a penalty term in the loss. Both [1] and [2] introduce gradient information as penalty terms for offline or online fine-tuning in reinforcement learning, but are only applicable to a single setting. In contrast, VDT leverages advantage-weighted and penalty losses, together with parallel sampling techniques, and is the first to achieve competitive performance in both offline and online fine-tuning scenarios. We will further elaborate on the comparison between VDT and [1][2] in the related work section. In addition, the table below presents a performance comparison of the methods, where it can be seen that VDT consistently achieves strong results across different settings.
> offline|CGDT|VDT
> -|-|-
> halfcheetah-medium|43.0|43.9
> hopper-medium|96.9|98.3
> walker2d-medium|79.1|81.6
> halfcheetah-medium-expert|93.6|93.9
> hopper-medium-expert|107.6|111.5
> walker2d-medium-expert|109.3|110.4
> average score|88.2|89.9
>
> online|TD3+ODT|VDT
> -|-|-
> halfcheetah-medium|66.9|53.5
> hopper-medium|89.1|108.1
> walker2d-medium|90.8|89.8
> halfcheetah-medium-expert|99.8|101.7
> hopper-medium-expert|112.6|117.8
> walker2d-medium-expert|111.9|112.7
> average score|95.1|97.3
>
> **W4: The proposed method seems to be sensitive with respect to $\epsilon$.**
> In Figure 2, we tested four values of $\epsilon$: 0.3, 0.5, 0.7, and 0.9. When $\epsilon$ is greater than 0.5, the value function tends to optimistically estimate the value of actions, while for $\epsilon$ less than 0.5, the value function tends to be pessimistic. Since the range of values used in our ablation study was relatively narrow, it was not easy to fully assess the model's sensitivity to $\epsilon$. Therefore, we further extended the experiments in Figure 2 by expanding the range of $\epsilon$ to [0.2, 0.3, 0.4, 0.5, 0.6, 0.7, 0.8, 0.9]. The table below shows the results of varying $\epsilon$ on both offline and online experiments. We found that the model's performance is relatively stable and optimal when $\epsilon$ is in the range of approximately 0.6 to 0.8. This observation is also generally consistent with the findings in the original IQL paper. Beyond a certain range, performance drops rapidly, which is related to the inherent properties of the value function. When $\epsilon$ is less than 0.5, the value function tends to underestimate values, making it difficult to effectively guide the model to achieve performance beyond the expert data. The same issue also arises when $\epsilon$ is set too high.
> Env|$\epsilon$|Offline|Online
> -|-|-|-
> hopper-medium-v2|0.2|43.8|83.6
> ||0.3|34.6|91.6
> ||0.4|32.0|85.0
> ||0.5|35.3|80.8
> ||0.6|75.8|105.0
> ||0.7|98.3|108.1
> ||0.8|96.0|106.0
> ||0.9|84.7|40.6
> |walker2d-medium-replay-v2|0.2|0.3|8.9
> ||0.3|0.5|18.9
> ||0.4|0.4|12.0
> ||0.5|5.6|10.6
> ||0.6|65.3|88.9
> ||0.7|82.3|95.5
> ||0.8|81.0|92.0
> ||0.9|19.7|79.4
>
> **MW1: On page 25, a space is missing.**
> We apologize for this typesetting error. We have reinserted the missing space and updated it in the subsequent version.
>
> **MW2: the subscripts are out of the circle and almost overlap with other parts in Fig1**
> We have updated Figure 1 by enlarging the circles to accommodate the subscripts and slightly adjusting the layout of longer subscripts to prevent the circles from becoming excessively large. As images cannot be directly displayed during the rebuttal period, we will present the updated figure in the main text in subsequent versions.
>
> **Q1: why is IQL taking comparable amount of parameters to decision transformer-based method and is much slower during online finetuning?**
> We apologize for this potentially confusing point. To evaluate performance, we replaced the policy network in the IQL algorithm with a transformer architecture whose prediction head outputs probabilities. This led to a rapid increase in the number of parameters and a noticeable decrease in convergence speed, although the performance remained comparable to the original IQL. We aimed to demonstrate that simply replacing the policy architecture in IQL with a transformer does not achieve the same effect as VDT. While the original IQL algorithm has a lower computational cost compared to VDT, this inevitably results in a loss of performance. To avoid such misunderstandings, we will revise the relevant table to include the original IQL algorithm with an MLP architecture, as shown in the table below.
> ||Memory↓|Params↑|ClockTime↓|Memory↓|Params↑|ClockTime↓
> -|-|-|-|-|-|-
> ||**Offline Training**|||**Online Tuning**||
> Original IQL|960M|0.6M|≈1.0h|960M|0.6M|≈1.0h
> IQL (Transformer policy)|2128M|3.31M|≈6.0h|2128M|3.31M|≈10.0h
> ODT|3968M|5.01M|≈9.0h|3968M|5.01M|≈4.5h
> VDT|4024M|5.24M|≈5.0h|4024M|5.24M|≈4.0h
>
> **L1: lack of evaluation on image-based environments**
> We thank the reviewers for this suggestion. We have evaluated the performance of VDT on the image-based Atari dataset, reporting the average performance across three random seeds, as shown in the table below. VDT also demonstrates competitive performance on the Atari dataset, indicating that our method is well-suited for both text and image-based datasets. This success was expected, given the precedents set by DT.
> Game|CQL|DT|DC|VDT
> -|-|-|-|-
> Breakout|211.1±15.2|242.4±31.8|352.7±44.7|420.8±29.4
> Qbert|104.2±8.7|28.8±10.3|67.0±14.7|69.4±5.0
> Pong|111.9±3.1|105.6±2.9|106.5±2.0|113.9±1.5
> Seaquest|1.7±0.2|2.7±0.4|2.6±0.3|3.9±0.7
> Asterix|4.6±0.5|5.2±1.2|6.5±1.0|8.7±1.1
> Frostbite|9.4±1.0|25.6±2.1|27.8±3.7|28.9±0.6
> Gopher|2.8±0.9|34.8±10.0|52.5±9.3|55.3±0.8

---

> > ### Comment · Reviewer_8M1K · 2025-08-01
> >
> > Thanks for the author's response. I particularly appreciate that the authors can provide experiment results on image-based environments, as well as detailed ablation results shown in the rebuttal. I think the reason from preventing me improving the score is: according to the authors' response, the main result of this model requires access to an accurate environment simulator, which is inherently advantageous to the baselines. Indeed, one can build world models / dynamic models in real-life applications; however, such models retrieved from transition data are not necessarily accurate, and the robustness of the method against inaccurate dynamic model is unknown.

---

> > > ### Author Response · Authors · 2025-08-02
> > >
> > > We sincerely appreciate the reviewer's recognition and fully understand the reviewer's concerns. In response, we conducted additional experiments where VDT directly uses the action predicted from the current state $\left(s_t^k, a_t^k\right)$ during sampling, without an environment simulator, to compute the Q-value (with Evaluation Horizon = 0). Our results show that even without multi-horizon value estimation, our model still achieves optimal or near-optimal performance on most tasks in both offline and online settings, and maintains a significant advantage over baseline models.
> > >
> > > |Offline|DT|DC|VDT (Evaluation Horizon=0)
> > > -|-|-|-
> > > halfcheetah-medium-replay|36.6|41.3|39.2
> > > hopper-medium-replay|82.7|94.2|95.3
> > > walker2d-medium-replay|79.4|76.6|82.3
> > > halfcheetah-medium|42.6|43.0|42.9
> > > hopper-medium|67.6|92.5|96.8
> > > walker2d-medium|74.0|79.2|80.0
> > > halfcheetah-medium-expert|86.8|93.0|93.6
> > > hopper-medium-expert|107.6|110.4|111.2
> > > walker2d-medium-expert|108.1|109.6|
> > >
> > > |Online|CQL|ODT|VDT (Evaluation Horizon=0)
> > > -|-|-|-
> > > halfcheetah-medium-replay|44.3|40.4|48.3
> > > hopper-medium-replay|95.3|88.9|117.0
> > > walker2d-medium-replay|78.0|76.9|92.9
> > > halfcheetah-medium|29.1|42.2|55.3
> > > hopper-medium|95.7|97.5|100.9
> > > walker2d-medium|89.4|76.8|88.4
> > > halfcheetah-medium-expert|99.9|100.9|101.8
> > > hopper-medium-expert|106.3|99.1|115.9
> > > walker2d-medium-expert|110.1|78.7|113.7
> > >
> > > To investigate the feasibility of constructing an environment simulator and to assess whether our approach can be adapted to dynamic models trained from offline data, we augmented VDT with additional state and reward prediction heads. This allows VDT to be trained directly on transition data, maximizing compatibility with the existing architecture and leveraging its inherent advantages. The experimental hyperparameters and pipeline remain largely unchanged, and we refer to this improved version as VDT-H. During the sampling process, given a pre-specified Rtg and initial environment state, we can autoregressively predict subsequent rewards, states, and actions to generate a simulated trajectory for value estimation. In addition, following previous works, we implemented a conceptually simple dynamics model using a standard six-layer MLP network, which takes state-action pairs as input and outputs the next state and reward through two separate prediction heads. This dynamic model is relatively easy to train based on MSE loss. It operates independently of VDT and provides state and reward predictions during the sampling process. Using such an independent dynamic model to assist VDT in sampling is denoted as VDT-I.
> > >
> > > The experimental results are as follows. Surprisingly, VDT-H continues to exhibit strong performance. We hypothesize that this is because the DT-based architecture for environment dynamics is better able to capture temporal dependencies and higher-order features present in offline data, thereby improving the accuracy of environment transition and reward modeling. In contrast, VDT-I generally underperforms VDT, possibly due to inaccuracies in dynamics prediction leading to erroneous value estimates and thus suboptimal decision-making. These results suggest that even in the absence of an accurate pre-existing dynamics model, it is still feasible to construct a dynamics model from offline data for trajectory simulation, and further demonstrate the robustness of VDT's sampling process to the quality of the environment simulator. We will provide the implementation details of both approaches in the appendix.
> > > ||DC|VDT|VDT-I|VDT-H
> > > -|-|-|-|-
> > > halfcheetah-medium-replay|41.3|39.4|38.1|39.0
> > > hopper-medium-replay|94.2|96.0|94.4|96.8
> > > walker2d-medium-repla|76.6|82.3|72.5|83.0
> > > halfcheetah-medium|43.0|43.9|44.8|43.9
> > > hopper-medium|92.5|98.3|96.8|97.8
> > > walker2d-medium|79.2|81.6|77.6|80.7
> > > halfcheetah-medium-expert|93.0|93.9|93.6|93.6
> > > hopper-medium-expert|110.4|111.5|109.2|111.7
> > > walker2d-medium-expert|109.6|110.4|105.8|110.0

---

> > > > ### Comment · Reviewer_8M1K · 2025-08-02
> > > >
> > > > Thanks for the author's detailed response; I am impressed by the timely and detailed experiment results provided by the authors. I do not have further concern and will increase my score from 4 to 5.

---

> > > > > ### Author Response · Authors · 2025-08-03
> > > > >
> > > > > We sincerely thank the reviewer for kindly raising the score. We will incorporate all new results obtained during the rebuttal period into the revised manuscript.

---

### Official Review · Reviewer_a7nS · 2025-06-23

**Clarity:** 2
**Significance:** 3
**Originality:** 2
**Rating:** 5
**Confidence:** 4

**Summary:**

The paper extends the Decision Transformer (DT) architecture by integrating value-based guidance into it. The approach is applicable both for offline learning and for offline->online setups (where one first trains offline, then fine-tunes). In the offline learning part, the approach uses value functions (coupled with an advantage calculation) to guide the DT. In the online phase, the approach uses a replay buffer and the value function to fine-tune the approach. The evaluation shows improvement over the Online DT baseline, as well as other approaches.

**Questions:**

The authors are welcome to address my comments above.

**Ethical Concerns:**

["NO or VERY MINOR ethics concerns only"]

**Final Justification:**

The authors' response addressed most of my concerns. I have updated the score.

**Limitations:**

YEs

**Paper Formatting Concerns:**

No concerns

**Quality:**

3

**Strengths And Weaknesses:**

Strengths:
1) Advantage-Weighted Learning and regularization - while this approach is not the first to incorporate value functions into DT (e.g., CGDT, which is a baseline), the use of advantage functions and learning is a novelty.
2) Effective offline-to-onilne setup - the proposed process has novel aspects. In addition, it seems to be (relatively) efficient and effective.
3) Robust evaluation: the appraoch is evaluated on a large number of baselines.

Weaknesses:
1) Evaluation - while the evaluation consists of multiple baselines, the authors only include the standard deviation for their approach. There are several "close calls" where seeing the standard deviation of the baselines would have been very helpful in assessing the contribution of the approach.
2) Relevant baselines - while CGDT is a very relevant baseline, a comparison to additional DT-based baselines, especially ones that are known to be good at stitching (Elastic DT is a strong candidate in my opinion, also because it also evaluated multiple options) could be helpful. Such a comparison would enable the readers to understand how the appraoch stands in comparison to other enhancements of DT.

---

> ### Author Rebuttal · Authors · 2025-07-31
>
> Thank you very much for your careful review of our work. We'll answer your questions one by one in the following, including some misunderstandings and some essential academic questions worth exploring.
>
> **W1: Seeing the standard deviation of the baselines would have been very helpful in assessing the contribution of the approach.**
> We sincerely thank the reviewers for their suggestions. We agree that standard deviations are important for reflecting the stability of each method and can further highlight the advantages of VDT over other approaches. In response, we have now added the standard deviations for all methods in Tables 1 and 2. Some of these values are directly taken from the original papers. For cases where the original papers did not report standard deviations, we reproduced the results using the available open-source code. We calculated the average standard deviation over the same three random seeds as VDT (123, 321, 132). The updated Tables 1 and 2 with standard deviations are shown below. As can be seen, VDT demonstrates better stability compared to most baseline methods, which further illustrates the advantages of our approach. Moreover, as the only method that achieves optimal performance in both offline and online settings, we believe that the VDT pipeline can also provide valuable insights for the offline RL community.
> ## Offline
> Gym Tasks|BEAR|BCQ|CQL|IQL|MoRel|BC|DT|StAR|GDT|CGDT|DC|VDT
> -|-|-|-|-|-|-|-|-|-|-|-|-
> halfcheetah-medium-replay-v2|38.6±2.1|34.8±1.7|37.5±0.3|**44.1**±0.5|40.2±2.0|36.6±2.33|36.6±1.6|36.8±1.8|40.5±2.2|40.4±2.0|41.3±3.8|39.4±2.0
> hopper-medium-replay-v2|33.7±2.0|31.1±1.6|95.0±5.2|92.1±8.1|93.6±4.7|18.1±2.07|82.7±3.9|29.2±1.5|85.3±4.3|93.4±4.7|94.2±5.9|**96.0**±1.9
> walker2d-medium-replay-v2|19.2±1.2|13.7±0.8|77.2±13.2|73.7±7.0|49.8±3.2|32.3±10.15|79.4±11.2|39.8±2.0|77.5±5.4|78.1±5.6|76.6±1.5|**82.3**±2.1
> halfcheetah-medium-v2|41.7±2.1|41.5±2.1|44.0±0.2|**47.4**±0.3|42.1±2.1|42.6±0.19|42.6±2.2|42.9±2.1|42.9±2.1|43.0±2.2|43.0±5.6|43.9±0.7
> hopper-medium-v2|52.1±2.6|65.1±3.3|58.5±3.77|63.8±5.7|95.4±4.8|52.9±1.76|67.6±2.2|59.5±3.0|77.1±3.9|96.9±4.8|92.5±0.9|**98.3**±0.1
> walker2d-medium-v2|59.1±3.0|52.0±2.6|72.5±3.28|79.9±1.8|77.8±3.9|75.3±16.24|74.0±1.2|73.8±3.7|76.5±3.8|79.1±4.0|79.2±0.6|**81.6**±1.7
> halfcheetah-medium-expert-v2|53.4±2.7|69.6±3.5|91.6±0.42|86.7±2.5|53.3±2.7|55.2±7.35|86.8±1.4|93.7±4.7|93.2±4.7|93.6±4.7|93.0±2.4|**93.9**±0.1
> hopper-medium-expert-v2|96.3±4.8|109.1±5.5|105.4±10.91|91.5±10.0|108.7±5.4|52.5±4.01|107.6±1.9|111.1±5.6|111.1±5.6|107.6±5.4|110.4±6.0|**111.5**±3.8
> walker2d-medium-expert-v2|40.1±2.0|67.3±3.4|108.8±0.39|109.6±0.8|95.6±4.8|107.5±15.98|108.1±0.3|109.0±5.5|107.7±5.4|109.3±5.5|109.6±1.1|**110.4**±0.9
>
> Adroit Tasks|BEAR|BCQ|CQL|IQL|MoRel|EDAC|BC|DT|D-QL|StAR|GDT|VDT
> -|-|-|-|-|-|-|-|-|-|-|-|-
> pen-human-v1|-1.0±0.2|66.9±6.7|37.5±3.8|71.5±28.5|-3.2±0.3|52.1±5.2|63.9±6.4|79.5±2.1|72.8±7.3|77.9±7.8|92.5±3.6|**126.7**±4.3
> hammer-human-v1|2.7±0.3|0.9±0.1|4.4±0.4|1.4±0.2|2.3±0.2|0.8±0.1|1.2±0.1|3.7±0.4|0.2±0.02|3.7±0.4|**5.5**±0.9|3.2±0.3
> door-human-v1|2.2±0.2|-0.05±0.01|9.9±1.0|4.3±0.4|2.3±0.2|10.7±1.1|2.0±0.2|14.8±1.5|0.0±0.0|1.5±0.2|18.6±2.9|**19.7**±0.5
> pen-cloned-v1|-0.2±0.02|50.9±5.1|39.2±3.9|37.3±29.9|-0.2±0.02|68.2±6.8|37.0±3.7|75.8±3.3|57.3±5.7|33.1±3.3|86.2±2.5|**145.6**±4.0
> hammer-cloned-v1|2.3±0.2|0.4±0.04|2.1±0.2|2.1±0.2|2.3±0.2|0.3±0.03|0.6±0.06|3.0±0.3|3.1±0.3|0.3±0.03|8.9±1.5|**19.6**±1.6
> door-cloned-v1|2.3±0.2|0.01±0.001|0.4±0.04|1.6±0.2|2.3±0.2|9.6±1.0|0.0±0.0|16.3±1.6|0.0±0.0|0.0±0.0|19.8±1.7|**30.6**±0.7
>
> Kitchen Tasks|BEAR|BCQ|CQL|IQL|O-RL|BC|DT|DD|StAR|GDT|DC|VDT
> -|-|-|-|-|-|-|-|-|-|-|-|-
> kitchen-complete-v0|0.0±0.0|8.1±0.4|43.8±2.2|62.5±3.1|2.0±0.1|65.0±3.3|50.8±2.5|65.0±3.3|40.8±2.0|43.8±2.2|40.9±0.3|**65.9**±0.2
> kitchen-partial-v0|13.1±0.7|18.9±0.9|49.8±2.5|46.3±2.3|35.5±1.8|33.8±1.7|57.9±2.9|57.0±2.9|12.3±0.6|73.3±3.7|66.8±5.9|**76.1**±10.8
>
> Maze2D Tasks|BEAR|BCQ|CQL|IQL|COMBO|BC|MPPI|DT|QDT|GDT|DC|VDT
> -|-|-|-|-|-|-|-|-|-|-|-|-
> maze2d-umaze-v1|65.7±3.3|49.1±2.5|86.7±4.3|42.1±2.1|76.4±3.8|85.7±4.3|33.2±1.7|31.0±1.6|57.3±2.9|50.4±2.5|20.1±6.9|**88.0**±4.6
> maze2d-medium-v1|25.0±1.3|17.1±0.9|41.8±2.1|34.9±1.7|38.5±1.9|38.3±1.9|10.2±0.5|8.2±0.4|13.3±0.7|7.8±0.4|38.2±2.4|**60.3**±0.5
>
> AntMaze Tasks|BEAR|BCQ|CQL|IQL|O-RL|BC|DT|RvS|StAR|GDT|DC|VDT
> -|-|-|-|-|-|-|-|-|-|-|-|-
> antmaze-umaze-v0|73.0±3.7|78.9±4.0|74.0±3.7|87.1±4.4|64.3±3.2|54.6±2.7|59.2±3.0|65.4±3.3|51.3±2.6|76.0±3.8|85.0±4.9|**100.0**±5.5
> antmaze-umaze-diverse-v0|61.0±3.1|55.0±2.8|84.0±4.2|64.4±3.2|60.7±3.0|45.6±2.3|66.2±3.3|60.9±3.0|45.6±2.3|69.0±3.5|78.5±9.7|**100.0**±4.7
> antmaze-medium-diverse-v0|8.0±0.4|0.0±0.0|53.7±2.7|**70.0**±3.5|0.0±0.0|0.0±0.0|7.5±0.4|67.3±3.4|0.0±0.0|0.0±0.0|0.0±0.8|30.0±2.8
>
> ## Online
> Dataset|TD3+BC|AWAC|CQL|IQL|PDT|ODT|VDT
> -|-|-|-|-|-|-|-
> halfcheetah-medium-replay-v2|48.1±1.2|39.0±2.1|44.3±1.5|44.0±0.3|42.8±1.0|40.4±1.6|**49.2**±1.3
> hopper-medium-replay-v2|90.7±3.5|79.6±4.2|95.3±2.8|93.5±4.4|94.8±3.1|88.9±6.3|**119.2**±5.7
> walker2d-medium-replay-v2|82.0±2.7|44.0±3.8|78.0±6.1|60.9±5.8|79.0±2.5|76.9±4.0|**95.5**±3.9
> halfcheetah-medium-v2|50.9±1.0|41.1±0.7|29.1±6.2|48.0±0.2|**69.5**±2.0|42.2±1.5|53.5±1.8
> hopper-medium-v2|64.6±2.2|91.0±3.1|95.7±2.5|44.3±4.1|100.2±2.7|97.5±2.1|**108.1**±4.2
> walker2d-medium-v2|85.2±5.0|79.1±8.8|89.4±2.3|68.9±2.3|88.1±2.6|76.8±8.3|**89.8**±2.7
> halfcheetah-medium-expert-v2|92.1±1.5|41.0±2.0|99.9±1.7|95.3±1.2|93.3±1.8|100.9±1.6|**101.7**±1.9
> hopper-medium-expert-v2|110.2±2.8|111.9±3.0|106.3±2.5|92.9±2.1|80.0±2.7|99.1±2.2|**117.8**±3.5
> walker2d-medium-expert-v2|110.1±7.6|78.3±7.9|110.1±9.4|109.6±0.2|108.9±2.5|78.7±2.8|**112.7**±2.9
> antmaze-umaze-v0|79.1±7.9|59.0±4.7|99.4±8.1|96.0±5.4|66.8±4.0|88.5±5.9|**110.0**±4.2
> antmaze-umaze-diverse-v0|78.1±3.9|49.0±3.2|99.4±7.2|84.0±6.4|79.3±3.7|56.0±5.7|**100.0**±5.9
> antmaze-medium-diverse-v0|56.7±2.8|0.3±0.1|32.3±4.0|72.0±8.9|63.4±3.5|55.6±6.2|**75.0**±2.7
>
> **W2: Relevant baselines**
> The core idea behind Elastic DT's ability to stitch trajectories lies in adaptive history truncation. During inference, EDT dynamically selects the optimal history length to maximize the maximum achievable return from the current state. Specifically, EDT estimates the maximum return for different history lengths using expected quantile regression and then chooses the history length to feed into the Transformer accordingly. EDT does not use a Q-function to guide action selection; instead, it essentially "forgets" unsuccessful histories, allowing the model to escape from low-return trajectories and stitch onto more optimal trajectory branches. We will further add a comparison of Elastic DT in both the methods and experiments sections of the main text. In addition, we have included a comparison with QDT [2], which combines the dynamic programming capabilities of Q-learning with the sequence modeling strengths of DT, enabling relabeling of return targets and thus improving DT's ability to stitch suboptimal trajectories.
>
> VDT differs in that it explicitly introduces a Q-function to provide value-guided weighting for action selection. At the same time, it employs multi-step Bellman equations and double Q-networks to stabilize value function training, thereby extending VDT to be applicable in both offline and online settings, and enabling parallel decision-making during sampling to leverage the evaluation capability of the Q-function fully. In the experimental section, as shown in the table below, we present a comparison of VDT with EDT and QCS. VDT demonstrates a clear advantage over EDT and QCS across a wide range of tasks.
>
> ||EDT|QCS|VDT
> -|-|-|-
> halfcheetah-medium|42.5|42.3|43.9
> hopper-medium|63.5|66.5|98.3
> walker2d-medium|72.8|67.1|81.6
> halfcheetah-medium-replay|37.8|35.6|39.4
> hopper-medium-replay|89.0|52.1|96.0
> walker2d-medium-replay|74.8|58.2|82.3
> average score|63.4|53.6|73.6
>
> [1] Elastic decision transformer, NeurIPS 2023.
> [2] Q-learning decision transformer: Leveraging dynamic programming for conditional sequence modelling in offline rl. PMLR 2023.

---

> > ### Comment · Reviewer_a7nS · 2025-08-05
> >
> > The authors' response addressed most of my concerns. I have updated the score.

---

> > > ### Author Response · Authors · 2025-08-05
> > >
> > > We sincerely thank the reviewer for kindly updating the score. We will incorporate all new results obtained during the rebuttal period into the revised manuscript.

---

### Official Review · Reviewer_x4AC · 2025-06-25

**Clarity:** 3
**Significance:** 2
**Originality:** 2
**Rating:** 4
**Confidence:** 3

**Summary:**

The paper proposes Value-Guided Decision Transformer (VDT), such that it integrates value-based learning into Decision Transformer (DT) by incorporating **advantage-weighting** on behavior cloning (BC) during offline training. Also, VDT proposes to unify offline training with online fine-tuning with trajectory-level replay buffers and return-to-go (RTG) alignment leading to robust performance. During action selection, Q-function based sampling process is used to choose more optimal actions.

**Questions:**

1. I am curious about the ablation study on how performance improves as we increase the range of RTGs together with Q-value based sampling, e.g. are we able to query higher RTGs than that of in the training dataset?

2. In online tuning stage, how much the performance depends on the initial offline training? how about ablating the different data ratios? how does VDT perform if the offline dataset has very poor coverage while including high-value trajectories?

**Ethical Concerns:**

["NO or VERY MINOR ethics concerns only"]

**Final Justification:**

The further experiments answered my questions regarding:
* Whether expanding RTG at inference will further enhance the performance: YES
* Ablations on various data ratios does illustrate the performance differences.

**Limitations:**

- As the authors also discussed, one primary limitation of the proposed method is the computational overhead of the Q-value guided sampling process during inference, especially when scaling to larger-scale, complex environments.

**Quality:**

2

**Strengths And Weaknesses:**

### Strengths

1. **Integrate Value-based Methods**: The integration of advantage-weighted learning and Q-function-based sampling is promising and bridging the gap between DT and other offline RL methods.

2. **Empirical Strength**: VDT achieves on-par or better results across diverse benchmarks, including Adroit, Kitchen, Maze2D, and AntMaze.


### Weaknesses

1. **Computational Overhead in Sampling**: Although possibly alleviated by GPU parallelization, the parallel sampling with multiple RTGs and evaluation horizons could become unaffordable for larger scale environments. Further development trading off between optimality and efficiency needs to be addressed.

---

> ### Author Rebuttal · Authors · 2025-07-31
>
> Thank you very much for your careful review of our work. We'll answer your questions one by one in the following, including some misunderstandings and some essential academic questions worth exploring.
>
> **W1: Computational Overhead in Sampling**
> The trade-off between exploration performance and computational cost is a common challenge faced by all deep learning methods. VDT requires parallel trajectory evaluation during sampling, which introduces some additional computational overhead. However, as shown in our experimental results in Table 4, this overhead is generally manageable due to efficient GPU batching. Compared to other methods, it does not become a bottleneck in terms of actual runtime or computational cost, while providing significant performance improvements. Therefore, we consider this additional overhead to be justified.
>
> **Q1: I am curious about the ablation study on how performance improves as we increase the range of RTGs together with Q-value based sampling, e.g. are we able to query higher RTGs than that of in the training dataset?**
> In our original work, we did not attempt to expand the range of RTGs mainly to avoid introducing excessive manual effort and to maintain the simplicity of our method. Intuitively, expanding the RTG range could further improve the algorithm's performance across different tasks. To investigate this, we conducted experiments in the offline setting on hopper-medium-v2, walker-medium-expert-v2, and pen-cloned-v1. In fact, during sampling on hopper-medium-v2, the maximum RTG used for querying the model in VDT already exceeded the maximum RTG in the training set, which is consistent with the usage in DT and aims to ensure fairness. However, we further explored whether expanding the RTG range could bring breakthroughs. For each task, we designed three additional RTG ranges, referred to as Ext. 1, Ext. 2, and Ext. 3.
>
> ---
> - **hopper-medium-v2** (max RTG in dataset: 3322)
>     - **Original range:** [7200, 3600, 1800]
>     - **Ext. 1:** [18000, 7200, 3600, 1800]
>     - **Ext. 2:** [72000, 36000, 18000, 7200, 3600, 1800, 720]
>     - **Ext. 3:** [72000, 36000, 18000, 7200, 3600, 1800]
> ---
> - **walker-medium-expert-v2** (max RTG: 5011)
>     - **Original range:** [5000, 4000, 2500]
>     - **Ext. 1:** [6000, 5000, 4000, 2500]
>     - **Ext. 2:** [8000, 6000, 5000, 4000, 2500, 1000]
>     - **Ext. 3:** [8000, 6000, 5000, 4000, 2500]
> ---
> - **pen-cloned-v1** (max RTG: 11271)
>     - **Original range:** [12000, 6000]
>     - **Ext. 1:** [15000, 12000, 6000]
>     - **Ext. 2:** [200000, 15000, 12000, 6000, 3000]
>     - **Ext. 3:** [200000, 15000, 12000, 6000]
> ---
>
> The purpose of Ext. 1 is to include higher RTGs, while Ext. 2 further increases the upper bound and adds a lowest RTG to provide a lower limit, aiming to maximize the RTG space available to the model without being excessive. Ext. 3 removes the lowest RTG to verify whether it affects sampling performance. To control variables, the evaluation horizon was fixed at 5. The experimental results demonstrate that expanding the RTG range consistently enhances performance, which implies that there is still room for improvement in the strategy used for RTG selection. In addition, the lowest RTG in the range has little impact on sampling performance, suggesting that the model may learn to utilize the largest possible RTG for decision-making without being affected by the lowest value.
> Task|Original Range|Ext. 1|Ext. 2|Ext. 3
> -|-|-|-|-
> hopper-medium-v2|98.3|99.1|101.6|102.0
> walker-medium-expert-v2|110.4|112.8|112.8|113.1
> pen-cloned-v1|145.6|149.5|148.3|149.2
>
> Furthermore, we observe that the RTG range and the evaluation horizon are highly correlated, and together they exhibit a synergistic effect that jointly influences the upper bound of performance during the sampling process. A possible explanation is that increasing the evaluation horizon allows the model to leverage better the guidance provided by RTGs. Conversely, if either the evaluation horizon or the RTG range is insufficiently large, this process may be hindered. We present the relevant experimental results on hopper-medium-v2.
> |Horizon|Original Range|Ext. 1|Ext. 2
> -|-|-|-
> 1|96.8|96.9|94.0
> 3|98.3|98.4|100.3
> 5|98.3|101.2|104.3
> 10|98.6|104.3|104.9
>
> **Q2: In online tuning stage, how much the performance depends on the initial offline training? how about ablating the different data ratios? how does VDT perform if the offline dataset has very poor coverage while including high-value trajectories?**
> We believe that the impact of offline training on online tuning can be analyzed from aspects such as the quality and quantity of offline data. For data quality, we have included datasets of varying quality within the same task. For example, in the hopper task, hopper-medium-replay represents the lowest quality, containing trajectories from random to medium policies; hopper-medium is of medium quality; and hopper-medium-expert is of the highest quality, consisting of expert policy trajectories. As shown in the experimental results in Table 1, the performance of VDT trained on datasets of different quality is highly correlated with the data quality.
>
> In the online tuning stage, we fine-tune VDT models that were pre-trained on offline datasets of varying quality. As shown in Table 2 of the main text, we observe that after fine-tuning, the performance of VDT pre-trained on different datasets is further improved, and the overall performance remains linearly correlated with the quality of the offline data utilized. However, in certain scenarios, the VDT pre-trained on the lowest quality dataset (hopper-medium-replay) outperforms those pre-trained on the higher quality datasets (hopper-medium and hopper-medium-expert). We hypothesize that this is mainly because under-trained models may be more sensitive to new gradient signals in some cases, making it easier for online fine-tuning to escape local optima and achieve greater improvements [1].
>
> [1] Levine S, Kumar A, Tucker G, et al. Offline reinforcement learning: Tutorial, review, and perspectives on open problems.
>
> To investigate the impact of offline data quantity on the final online fine-tuning performance, we further subsampled the task datasets in the Mujoco benchmark by randomly selecting trajectories and retaining only 5%, 10%, and 50% of the original data. Notably, the various Mujoco datasets contain an average of approximately 2,000 trajectories, which is relatively small compared to typical vision and language datasets. After subsampling, we can evaluate the performance of VDT under the most challenging data-scarce scenarios.
> It is worth noting that subsampled expert-level datasets can effectively represent scenarios with limited data coverage but containing high-value trajectories. This is because the significant reduction in data volume inevitably leads to lower coverage, while the inherent characteristics of expert datasets ensure the presence of sufficiently high-quality trajectories.
>
> As shown in the experimental results below, we observe that the performance degradation caused by reducing the data quantity is not substantial. When the data is reduced to only 5% of the original amount, the performance on most tasks remains at around 90% of the initial level. Furthermore, since the number of training steps is kept constant in our experiments, it is possible to further improve performance under reduced data conditions by increasing the number of training steps.
> Task|5%|10%|50%|100%
> -|-|-|-|-
> halfcheetah-medium-v2|49.8|53.6|52.1| 53.5
> hopper-medium-v2|102.9| 107.1|107.9|108.1
> walker-medium-v2|84.6|88.0|89.9|89.8
> halfcheetah-medium-expert-v2|80.3|97.6|101.9|101.7
> hopper-medium-expert-v2|109.6|110.0|118.7|117.8
> walker-medium-expert-v2|107.2|106.0|112.7|112.7
>
> We also investigated the impact of the number of offline training steps on the final performance of online tuning. In the original experiments, we used 10,000 offline pre-training steps, after which the model had almost converged. Keeping all other hyperparameters unchanged, we varied the number of offline pre-training steps to 1,000, 5,000, and 10,000 to simulate VDT models that had not fully converged during pre-training. The experimental results are shown in the table below. It can be seen that the effect of the number of offline training steps on final performance is highly similar to the effect of insufficient data on final performance. Both can be attributed to underfitting scenarios. As the number of offline pre-training steps increases, there is still room for performance improvement, which is consistent with intuition.
> Task|1000|5000|10000
> -|-|-|-
> halfcheetah-medium-v2|40.3|51.9|53.5
> hopper-medium-v2|88.6|100.2|108.1
> walker-medium-v2|33.1|76.0|89.8
> halfcheetah-medium-expert-v2|79.9|88.3|101.7
> hopper-medium-expert-v2|65.6|99.0|117.8
> walker-medium-expert-v2|48.0|101.5|112.7
>
> **L1: The proposed method is the computational overhead of the Q-value guided sampling process during inference.**
> In Table 4 of the main text, we compare the VDT algorithm with other methods. We observe that the average computational cost of VDT remains lower than that of the commonly used ODT, and this overhead is relatively manageable. When scaling VDT to large and complex environments, the cost can be further mitigated by adopting more efficient batching strategies or model pruning techniques. As discussed in Q1, compared to the potential performance gains brought by VDT through the integration of value functions and parallelized sampling, this additional overhead may be relatively acceptable. More importantly, our approach is currently the only method that demonstrates a significant performance advantage in both offline and online settings. We hope that our work can also provide the offline RL community with some inspiration for designing more general-purpose RL algorithms from a new perspective.

---

> > ### Comment · Reviewer_x4AC · 2025-08-08
> >
> > Thanks very much for addressing the questions and issues by conducting further experiments and ablation studies.
> > The empirical evidence indeed looks promising, accordingly, I would like to raise my score.

---

> > > ### Author Response · Authors · 2025-08-08
> > >
> > > We sincerely thank the reviewer for kindly raising the score. We will incorporate all new results obtained during the rebuttal period into the revised manuscript.

---

### Official Review · Reviewer_yr9t · 2025-07-02

**Clarity:** 2
**Significance:** 2
**Originality:** 3
**Rating:** 4
**Confidence:** 3

**Summary:**

This paper introduces value-function guidance into the decision transformer for trajectory sequence prediction (conditional-sequence-modeling) problems in reinforcement learning. This value-function guidance enables the model to (1) stitch together sub-optimal trajectories during offline training and (2) keep improving rapidly with only a few online interactions. Experiments show that the proposed value-guided decision transformers (VGDT) outperform prior methods in both offline and offline-to-online settings.

**Questions:**

- Line 153: Why is the expectile loss function an unbiased estimate of the conditional expectation? The IQL [1] paper has already shown that the value function will converge to the maximal Q-function within the support of the dataset.

- Line 160: Why do we prefer the n-step TD loss over the simpler one-step TD loss?

- Eq. 5: Any empirical evidence to show the effect and significance of $\eta$ and $\lambda$?

- Line 173 - 175: This sentence is confusing. How does this loss function optimize the stability of trajectory modeling? How is it able to prevent the local optima within the data distribution?

- Line 195 - 196: Why is the two-step sampling procedure uniform? From the sentence, it sounds like the method is prioritizing trajectory segments from longer trajectories.

- Line 199 - 206: In the online learning phase, does the method use the current ongoing trajectory to update the DT?

- The sampling process in Sec. 4.3 is similar to both the cross-entropy method in model-based RL [2] and the Selecting from Behavior Candidates (SfBC) [3] policy extraction method in offline RL. Including some discussions of these two categories of methods in this section could be helpful.

- Line 215: How do we predefine the m candidate RTG? Are they varying across different tasks?

- Fig 3: From this figure, it looks like the effect of the evaluation horizon is minor – the performance for $E = 1$ and $E \geq 1$ is not significantly different. Then, why do we not prefer a simpler method?

[1] Kostrikov, Ilya, Ashvin Nair, and Sergey Levine. "Offline Reinforcement Learning with Implicit Q-Learning." In International Conference on Learning Representations.

[2] Chua, Kurtland, Roberto Calandra, Rowan McAllister, and Sergey Levine. "Deep reinforcement learning in a handful of trials using probabilistic dynamics models." Advances in neural information processing systems 31 (2018).

[3] Chen, Huayu, Cheng Lu, Chengyang Ying, Hang Su, and Jun Zhu. "Offline Reinforcement Learning via High-Fidelity Generative Behavior Modeling." In The Eleventh International Conference on Learning Representations.

**Ethical Concerns:**

["NO or VERY MINOR ethics concerns only"]

**Final Justification:**

The presented theoretical characterization of the gap between one-step and optimal value functions is helpful, the hyperparameter ablations substantiate the proposed components, and the image-based experiments further attest to VDT’s effectiveness. Accordingly, I will raise my rating to 4.

**Limitations:**

The limitation section only discussed the computation cost of the algorithm. Including some discussions about the theoretical issue or missing properties of the current method could be helpful.

**Paper Formatting Concerns:**

N / A

**Quality:**

3

**Strengths And Weaknesses:**

**Strengths**

- This paper uses value-function guidance to unify the learning of the decision transformer in both offline and online RL settings. By leveraging the Q-function learned by an n-step implicit Q learning loss, the model is able to use an advantage-weighted regression loss to predict the action sequences, enabling stitching from sub-optimal trajectories and value-guided action selection during inference.

- Through experiments on D4RL benchmarks, the paper shows that VGDT achieves competitive empirical performance, compared to prior offline reinforcement learning methods and other decision transformer variants. Additional results ablate different components of the algorithm and show the effects of them in the complete VGDT.

**Weaknesses**

- Theorem 4.1 seems to state that the optimal decision transformer derived from the proposed method results in *one-step* policy improvement, which does not provide a guarantee to converge to the optimal policy. In practice, there has been success in using *one-step* policy improvement to derive an RL method. But some connections between the one-step value function and the optimal value function will still be helpful.

- For the experiments, ablation studies of some hyperparameters are missing. Also, including some results on benchmarks other than D4RL or tasks with image-based observations can potentially make the experiment more convincing.

---

> ### Author Rebuttal · Authors · 2025-07-31
>
> Thanks for reviewing our work attentively. We'll answer your questions in the following.
>
> **W1: Connections between the one-step value function and the optimal value function will still be helpful.**
> We can indeed provide further theoretical guarantees for the strong empirical performance of VDT. To this end, we propose Theorem 2, which establishes an upper bound for the one-step value function and quantifies its difference from the optimal value function.
>
> **Theorem 2.** For any initial state distribution $\mu$, we have
>
> $$V^{ \pi^* }( \mu )-V^{ \pi_{DT}^* }( \mu) \leq \frac{ 2 \gamma }{(1- \gamma )^2} \cdot \mathbb{ E }_{ s \sim d^{ \pi^* }}\left[ \max _{ a \notin \mathcal{ A }_D(s)} Q^{ \pi^* }(s, a)- \max _{a \in  \operatorname{ supp }( \beta( \cdot \mid s)) } Q^{\pi^*}(s, a)\right ]$$
>
> Here, $\mathcal{A}_D(s)$ denotes the action support of the dataset at state $s$, the set $\mathrm{supp}(\beta(\cdot \mid s))$ denotes the set of actions observed in the dataset at state $s$. Due to the strict word limit during the rebuttal phase, we will include the detailed proof in the appendix. Theorem 2 complements Theorem 1 by further elucidating the quantitative relationship between VDT and the optimal policy $\pi^*$. When the dataset has good coverage, our policy is nearly optimal; even with limited coverage, VDT stays conservatively optimal within the supported regions.
>
> **W2&Q3: Empirical evidence of hyperparameterss $\eta$ and $\lambda$ and image-based observations.**
> We have already conducted an experimental analysis of the range of $\eta$ in Table 9 of the Appendix E. Additionally, we have included an ablation study on $\lambda$ in the offline setting. We found that the value of $\lambda$ does not significantly affect the experimental results (except $\lambda = 0$). This further demonstrates that VDT is highly robust to the choice of $\lambda$.
> Game|$\lambda$=0|$\lambda$=0.3|$\lambda$=0.5|$\lambda$=1
> -|-|-|-|-
> hopper-medium|89.0|96.9|98.3|98.4
> walker2d-medium-expert|100.9|110.4|110.4|109.5
> antmaze-umaze|84.5|100.0|100.0|98.0
>
> We evaluated VDT on the Atari dataset with image-based observations, averaging results over three random seeds. Compared to existing methods, VDT achieves superior performance, demonstrating its generalization ability across diverse tasks.
> Game|CQL|DT|DC|VDT
> -|-|-|-|-
> Breakout|211.1±15.2|242.4±31.8|352.7±44.7|420.8±29.4
> Qbert|104.2±8.7|28.8±10.3|67.0±14.7|69.4±5.0
> Pong|111.9±3.1|105.6±2.9|106.5±2.0|113.9±1.5
> Seaquest|1.7±0.2|2.7±0.4|2.6±0.3|3.9±0.7
> Asterix|4.6±0.5|5.2±1.2|6.5±1.0|8.7±1.1
> Frostbite|9.4±1.0|25.6±2.1|27.8±3.7|28.9±0.6
> Gopher|2.8±0.9|34.8±10.0|52.5±9.3|55.3±0.8
>
> **Q1: Why is the expectile loss function an unbiased estimate of the conditional expectation?**
> We apologize for our spelling mistake. The expectile loss function is an unbiased estimator of the conditional expectile, not the conditional expectation, except when the expectile parameter $\tau = 0.5$.
>
> **Q2: Why do we prefer the n-step TD loss over the simpler one-step TD loss?**
> We use n-step TD loss instead of one-step TD loss because n-step targets better balance bias and variance, leading to faster and more stable learning, especially with sparse or delayed rewards. Prior work has shown the benefits of n-step methods in offline RL [1][2][3]. Our ablation experiments confirm that n-step generally outperforms one-step, but using too many steps can also have negative effects. This is likely because longer returns amplify errors from distributional shift, offsetting the benefits of n-step learning.
> ||1-step|5-step|8-step
> -|-|-|-
> hopper-medium-v2|90.0|98.3|97.6
> walker2d-medium-expert-v2|87.2|110.4|113.8
> maze2d-medium|34.9|60.3|60.4
> antmaze-umaze|88.0|100.0|90.0
> average score|75.0|92.3|90.4
>
> [1] Reinforcement learning: An introduction.
> [2] Fixed-horizon temporal difference methods for stable RL.
> [3] Adaptive temporal-difference learning for policy evaluation with per-state uncertainty estimates.
>
> **Q4: How does this loss function optimize the stability of trajectory modeling and prevent the local optima.**
> These two advantages correspond to the two terms in the loss function. The advantage-weighted term promotes imitation of high-value actions, reducing the impact of low-value or outlier actions and improving stability—an idea also discussed in the IQL paper. The regularization term encourages the policy to explore actions with higher Q-values, even if they are rare in the data, helping the policy escape local optima within the data distribution. The results in Tables 1 and 2 empirically verify that VDT can indeed learn optimal actions that surpass most previous methods. Furthermore, the visualisation results in Tables 4 and 8 in the appendix also demonstrate, from the perspectives of convergence time and reward curves, that VDT exhibits a certain degree of competitiveness in terms of stability.
>
> **Q5: Why is the two-step sampling procedure uniform?**
> This is because longer trajectories contain more sub-sequences. By sampling trajectories in the first step with probability proportional to their length, and then uniformly selecting a sub-sequence from within each trajectory in the second step, every possible sub-sequence in the entire dataset has an equal probability of being chosen. In other words, the higher chance of picking a long trajectory is balanced out by the fact that there are more sub-sequences to choose from within it, resulting in uniform sampling across all sub-sequences.
>
> **Q6: In the online learning phase, does the method use the current ongoing trajectory to update the DT?**
> If "current ongoing trajectory" refers to the complete trajectory obtained after the model interacts with the environment once, then answer is yes. After each full episode is collected using the current policy, we perform return-to-go alignment on the trajectory and add it to the replay buffer. The DT is then updated using trajectories sampled from this replay buffer, which now includes the most recent trajectory collected from the environment. Therefore, during the online fine-tuning phase, we follow an interact-sample-update loop, which ensures that the model continually incorporates the latest experiences. The training procedure for the online fine-tuning phase is presented in Algorithms 1 and 2 in Appendix C.
>
> **Q7: Including some discussions of these two categories of methods in this section could be helpful.**
> We will add the following content to this section:
> "Our sampling process is similar to CEM [2] and SfBC [3]. Like CEM, we generate multiple candidate actions at each step, evaluate them with a value function over a short planning horizon, and select the best one—essentially a single-step population-based search. Unlike SfBC, which samples from a behavior policy and selects by Q-value, VDT uses RTGs to guide candidate actions and evaluates them in parallel. This integration of RTG guidance and batch evaluation allows VDT to combine the strengths of population search and candidate selection while maintaining efficient inference."
>
> **Q8: How do we predefine the m candidate RTG?**
> The candidate RTGs vary across different tasks, and the selection of RTGs is largely consistent with that of the DT. Our main focus is on utilizing candidate RTGs effectively, specifically leveraging the guidance provided by different candidate RTGs. As can be seen, in some tasks, we use only a small number of candidate RTGs, and we did not deliberately increase the number or variety of RTGs (although this could potentially lead to performance improvements). The primary reason for this is to ensure a fair comparison and to minimize the need for manual design. Due to space limitations, we present the candidate RTGs used in some tasks below. A more detailed table will be reintroduced in the main text.
> Environment|Targets
> -|-
> hopper|[7200,3600,1800]
> walker2d|[5000,4000,2500]
> pen|[12000,6000]
> hammer|[12000,6000,3000]
> door|[2000,1000,500]
> relocate|[3000,1000]
> maze2d|[300,200,150,100,50,20]
>
> **Q9: Fig 3 looks like the effect of the evaluation horizon is minor?**
> Since the evaluation horizon is decoupled from other innovations in VDT, it is indeed possible to omit this component for the sake of simplicity. However, we are still eager to explore opportunities for optimizing the sampling process. Although the performance improvements shown in Figure 3 are not substantial enough to be considered groundbreaking for the field (partly due to the strong baseline performance of the model itself), we believe this direction remains valuable. As mentioned in Q8, we did not carefully select the candidate RTGs. We further expanded the range of candidate RTGs used for hopper-medium, as shown in the table below. Surprisingly, we found that increasing the evaluation horizon leads to different changes in model performance across different RTG ranges. This suggests that effective collaboration between the evaluation horizon and RTG selection can potentially lead to further performance gains. Moreover, due to the parallelism in the evaluation process, we believe that the minor additional computational cost introduced by the evaluation horizon is entirely acceptable.
> Horizon|[7200,3600,1800]|[18000,7200,3600,1800]|[72000,36000,18000,7200,3600,1800,720]
> -|-|-|-
> 1|96.8|96.9|94.0
> 3|98.3|98.4|100.3
> 5|98.3|101.2|104.3
> 10|98.6|104.3|104.9
>
> **L1: Including some discussions about the theoretical issue or missing properties could be helpful.**
> We have reintroduced the analysis of the gap between the single-step value function used in VDT and the optimal value function, providing an upper bound and further improving the theoretical framework of VDT. In addition, following the valuable suggestions from the reviewers, we have refined many details of existing methods, including ablation studies on components and hyperparameters, literature comparisons, and discussions of potential advantages. All of these improvements will be incorporated into the revised manuscript.

---

> > ### Comment · Reviewer_yr9t · 2025-08-01
> >
> > I appreciate the comprehensive response. The presented theoretical characterization of the gap between one-step and optimal value functions is helpful, the hyperparameter ablations substantiate the proposed components, and the image-based experiments further attest to VDT’s effectiveness. I recommend incorporating these results into the revised version. Accordingly, I will raise my rating to 4.

---

> > > ### Author Response · Authors · 2025-08-02
> > >
> > > We sincerely thank the reviewer for kindly raising the score. We will incorporate all these results into the revised version as suggested.

---

### Note · Authors · 2025-08-14

We would like to express our deepest gratitude to the Program Chairs, Senior Area Chairs, and Area Chairs for their valuable guidance and support throughout this process. We sincerely thank the reviewers for their constructive feedback and thoughtful suggestions. We are encouraged by the reviewers’ appreciation that

**The design of our method is novel, engaging, and well-motivated**

- "The integration of advantage-weighted learning and Q-function-based sampling is promising and bridging the gap between DT and other methods." - x4AC

- "The use of advantage functions and learning is a novelty and the proposed process has novel aspects." - a7nS

- "The idea is intuitive and it is very interesting to see the idea of sampling from different RTGs with parallel trajectories which tries to address a crucial concern of DT." - 8M1K

**Our paper is neat, well-written, and easy to follow**

- "The paper is well-written and very easy to follow; the design are clearly conveyed." - 8M1K

- "It is very well-rounded in the sense that it provides detailed hyperparameters, environment specifications and pseudocode that helps the understanding of VDT." - 8M1K

**Experiments and ablation studies are thorough and detailed, showing substantial improvements**

- "The VDT achieves competitive empirical performance, compared to prior offline RL methods and other DT variants." - yr9t

- "VDT achieves on-par or better results across diverse benchmarks." - x4AC

- "The appraoch is evaluated on a large number of baselines and it seems to be efficient and effective." - a7nS

- "The experiment results seems quite solid, which I feel sufficient for proving the effectiveness of the method." - 8M1K

During the rebuttal and discussion phases, we thoroughly addressed all issues raised by the reviewers with extensive additional experiments. All reviewers expressed satisfaction with our responses and increased their ratings. Some of the key improvements include providing new theoretical analysis and proofs regarding optimality and convergence (yr9t); adding comprehensive ablation studies and validating generalization on image-based environments (yr9t, x4AC, 8M1K); demonstrating the robustness of VDT (8M1K); expanding baseline comparisons and reporting standard deviations for all methods, as well as including baselines such as Elastic DT (a7nS); and clarifying the efficiency and scalability of our sampling scheme (yr9t, x4AC). All improvements will be included in the revised paper.

---

### Decision · Program_Chairs · 2025-09-17

**Decision:**

Accept (poster)

**Comment:**

This paper incorporates guidance from value functions into the decision transformer framework for offline learning, and for online tuning of offline-trained policies.  In empirical evaluations, the approach is competitive (either matching or outperforming) a large variety of baselines.  All of the reviewers found the experiments to be both convincing and extensive; one reviewer did suggest an additional baseline (Elastic DT), which the authors will include in the final revision.

Overall, the technique appears to be both well motivated and empirically well performing; I think it is likely to have significant impact.